# AN EFFICIENT AND MARGIN-APPROACHING ZERO-CONFIDENCE ADVERSARIAL ATTACK

## ABSTRACT

There are two major paradigms of white-box adversarial attacks that attempt to impose input perturbations. The first paradigm, called the fix-perturbation attack, crafts adversarial samples within a given perturbation level. The second paradigm, called the zero-confidence attack, finds the smallest perturbation needed to cause misclassification, also known as the margin of an input feature. While the former paradigm is well-resolved, the latter is not. Existing zero-confidence attacks either introduce significant approximation errors, or are too time-consuming. We therefore propose MARGINATTACK, a zero-confidence attack framework that is able to compute the margin with improved accuracy and efficiency. Our experiments show that MARGINATTACK is able to compute a smaller margin than the state-of-the-art zero-confidence attacks, and matches the state-of-the-art fix-perturbation attacks. In addition, it runs significantly faster than the Carlini-Wagner attack, currently the most accurate zero-confidence attack algorithm.

## 1 INTRODUCTION

Adversarial attack refers to the task of finding small and imperceptible input transformations that cause a neural network classifier to misclassify. White-box attacks are a subset of attacks that have access to gradient information of the target network. In this paper, we will focus on the white-box attacks. An important class of input transformations is adding small perturbations to the input. There are two major paradigms of adversarial attacks that attempt to impose input perturbations. The first paradigm, called the fix-perturbation attack, tries to find perturbations that are most likely to cause misclassification, with the constraint that the norm of the perturbations cannot exceed a given level. Since the perturbation level is fixed, fix-perturbation attacks may fail to find any adversarial samples for inputs that are far away from the decision boundary. The second paradigm, called the zero-confidence attack, tries to find the smallest perturbations that are guaranteed to cause misclassification, regardless of how large the perturbations are. Since they aim to minimize the perturbation norm, zero-confidence attacks usually find adversarial samples that ride right on the decision boundaries, and hence the name "zero-confidence". The resulting perturbation norm is also known as the *margin* of an input feature to the decision boundary. Both of these paradigms are essentially constrained optimization problems. The former has a simple convex constraint (perturbation norm), but a non-convex target (classification loss or logit differences). In contrast, the latter has a non-convex constraint (classification loss or logit differences), but a simple convex target (perturbation norm).

Despite their similarity as optimization problems, the two paradigms differ significantly in terms of difficulty. The fix-perturbation attack problem is easier. The state-of-the-art algorithms, including projected gradient descent (PGD) (Madry et al., 2017) and distributional adversarial attack (Zheng et al., 2018), can achieve both high efficiency and high success rate, and often come with theoretical convergence guarantee. On the other hand, the zero-confidence attack problem is much more challenging. Existing methods are either not strong enough or too slow. For example, DeepFool (Moosavi Dezfooli et al., 2016) and fast gradient sign method (FGSM) (Goodfellow et al., 2014; Kurakin et al., 2016a;b) linearizes the constraint, and solves the simplified optimization problem with a simple convex target and a linear constraint. However, due to the linearization approximation errors, the solution can be far from optimal. As another extreme, L-BFGS (Szegedy et al., 2013) and Carlini-Wagner (CW) (Carlini & Wagner, 2017) convert the optimization problem into a Lagrangian, and the Lagrangian multiplier is determined through grid search or binary search. These attacks are generally much stronger and theoretically grounded, but can be very slow.

The necessity of developing a better zero-confidence attack is evident. The zero-confidence attack paradigm is a more realistic attack setting. More importantly, it aims to measure the margin of each individual token, which lends more insight into the data distribution and adversarial robustness. Motivated by this, we propose MARGINATTACK, a zero-confidence attack framework that is able to compute the margin with improved accuracy and efficiency. Specifically, MARGINATTACK iterates between two moves. The first move, called restoration move, linearizes the constraint and solves the simplified optimization problem, just like DeepFool and FGSM; the second move, called projection move, explores even smaller perturbations without changing the constraint values significantly. By construction, MARGINATTACK inherits the efficiency in DeepFool and FGSM, and improves over them in terms of accuracy with a convergence guarantee. Our experiments show that MARGINAT-TACK attack is able to compute a smaller margin than the state-of-the-art zero-confidence attacks, and matches the state-of-the-art fix-perturbation attacks. In addition, it runs significantly faster than CW, and in some cases comparable to DeepFool and FGSM.

## 2 RELATED WORKS

In addition to the aforementioned state-of-the-art attacks, there are a couple of other works that attempt to explore the margin. Jacobian-based saliency map attack (Papernot et al., 2016) is among the earliest works that apply gradient information to guide the crafting of adversarial examples. It chooses to perturb the input features whose gradient is consistent with the adversarial goal. One-pixel attack (Su et al., 2017) finds adversarial examples by perturbing only one pixel, which can be regarded as finding the $\ell_0$ margin of the inputs. Ilyas et al. (2018) converts PGD into a zero-confidence attack by searching different perturbation levels, but this again can be time-consuming because it needs to solve multiple optimization subproblems. Weng *et al.* proposed a metric called CLEVER (Weng et al., 2018), which estimates an upper-bound of the margins. Unfortunately, recent work (Goodfellow, 2018) has shown that CLEVER can overestimate the margins due to gradient masking (Papernot et al., 2017). The above are a just a small subset of white-box attack algorithms that are relevant to our work. For an overview of the field, we refer readers to Akhtar & Mian (2018).

The MARGINATTACK framework is inspired by the Rosen's algorithm (Rosen, 1961) for constraint optimization problems. However, there are several important distinctions. First, the Rosen's algorithm rests on some unrealistic assumptions for neural networks, *e.g.* continuously differentiable constraints, while MARGINATTACK has a convergence guarantee with a more realistic set of assumptions. Second, the Rosen's algorithm requires a step size search for each iteration, which can be time-consuming, whereas MARGINATTACK will work with a simple diminishing step size scheme. Most importantly, as will be shown later, MARGINATTACK refers to a large class of attack algorithms depending on how the two parameters, $a^{(k)}$ and $b^{(k)}$, are set, and the Rosen's algorithm only fits into one of the settings, which only works well under the $\ell_2$ norm. For other norms, there exist other parameter settings that are much more effective. As another highlight, the convergence guarantee of MARGINATTACK holds for all the settings that satisfy some moderate assumptions.

## 3 THE MARGINATTACK ALGORITHM

In this section, we will formally introduce the algorithm and discuss its convergence properties. In the paper, we will denote scalars with non-bolded letters, *e.g.* $a$ or $A$; column vectors with lower-cased, bolded letters, *e.g.* $\boldsymbol{a}$; matrix with upper-cased, bolded letters, *e.g.* $\boldsymbol{A}$; sets with upper-cased double-stoke letters, *e.g.* $\mathbb{A}$; gradient of a function $f(\boldsymbol{x})$ evaluated at $\boldsymbol{x} = \boldsymbol{x}_0$ as $\nabla f(\boldsymbol{x}_0)$.

### 3.1 PROBLEM FORMULATION

Given a classifier whose output logits are denoted as $l_0(\boldsymbol{x}), l_1(\boldsymbol{x}), \cdots, l_{C-1}(\boldsymbol{x})$, where $C$ is the total number of classes, for any data token $(\boldsymbol{x}_0, t)$, where $\boldsymbol{x}_0$ is an $n$-dimensional input feature vector, and $t \in \{0, \cdots, C-1\}$ is its label, MARGINATTACK computes

$$\boldsymbol{x}^* = \arg\min_{\boldsymbol{x}} \ d(\boldsymbol{x} - \boldsymbol{x}_0), \ \text{s.t.} \ c(\boldsymbol{x}) \le 0, \tag{1}$$

where $d(\cdot)$ is a norm. In this paper we only consider $\ell_2$ and $\ell_\infty$ norms, but the proposed method is generalizable to other norms. For non-targeted adversarial attacks, the constraint is defined as

$$c(\boldsymbol{x}) = l_t(\boldsymbol{x}) - \max_{i \ne t} l_i(\boldsymbol{x}) - \varepsilon, \tag{2}$$

where $\varepsilon$ is the offset parameter. As a common practice, $\varepsilon$ is often set to a small negative number to ensure that the adversarial sample lies on the incorrect side of the decision boundary. In this paper, we will only consider non-targeted attack, but all the discussions are applicable to targeted attacks (i.e. $c(\boldsymbol{x}) = \max_{i \neq a} l_i(\boldsymbol{x}) - l_a(\boldsymbol{x}) - \varepsilon$ for a target class $a$).

## 3.2 THE MARGINATTACK PROCEDURE

MARGINATTACK alternately performs the restoration move and the projection move. Specifically, denote the solution after the $k$-th iteration as $\boldsymbol{x}^{(k)}$. Then the two steps are:

**Restoration Move**: The restoration move tries to hop to the constraint boundary, i.e. $c(\boldsymbol{x}) = 0$ with the shortest hop. Formally, it solves:

$$\boldsymbol{z}^{(k)} = \arg\min_{\boldsymbol{x}} d(\boldsymbol{x} - \boldsymbol{x}^{(k)}), \quad \text{s.t. } \nabla^T c(\boldsymbol{x}^{(k)})(\boldsymbol{x} - \boldsymbol{x}^{(k)}) = -\alpha^{(k)} c(\boldsymbol{x}^{(k)}). \quad (3)$$

where $\alpha^{(k)}$ is the step size within $[0, 1]$. Notice that the left hand side of the constraint in Eq. (3) is the first-order Taylor approximation of $c(\boldsymbol{z}^{(k)}) - c(\boldsymbol{x}^{(k)})$, so this constraint tries to move point closer to $c(\boldsymbol{x}) = 0$ by $\alpha^{(k)}$. It can be shown, from the dual-norm theory,[1] that the solution to (3) is

$$\boldsymbol{z}^{(k)} = \boldsymbol{x}^{(k)} - \frac{\alpha^{(k)} c(\boldsymbol{x}^{(k)}) \boldsymbol{s}(\boldsymbol{x}^{(k)})}{\nabla^T c(\boldsymbol{x}^{(k)}) \boldsymbol{s}(\boldsymbol{x}^{(k)})} \quad (4)$$

$\boldsymbol{s}(\boldsymbol{x})$ is defined such that $\nabla^T c(\boldsymbol{x}) \boldsymbol{s}(\boldsymbol{x}) = d^*(\nabla^T c(\boldsymbol{x}))$, where $d^*(\cdot)$ is the dual norm of $d(\cdot)$. Specifically, noticing that the dual norm of the $\ell_p$ norm is the $\ell_{(1-p^{-1})^{-1}}$ norm, we have

$$s(\boldsymbol{x}) = \begin{cases} \nabla c(\boldsymbol{x})/\|\nabla c(\boldsymbol{x})\|_2 & \text{if } d(\cdot) \text{ is the } \ell_2 \text{ norm} \\ \text{sign}(\nabla c(\boldsymbol{x})) & \text{if } d(\cdot) \text{ is the } \ell_\infty \text{ norm} \end{cases} \quad (5)$$

As mentioned, Eq. (4) is similar to DeepFool under $\ell_2$ norm, and to FGSM under $\ell_\infty$ norm. Therefore, we can expect that the restoration move should effectively hop towards the decision boundary, but the hop direction may not be optimal. That is why we need the next move.

**Projection Move**: The projection move tries to move closer to $\boldsymbol{x}_0$ while ensuring that $c(\boldsymbol{x})$ will not change drastically. Formally,

$$\boldsymbol{x}^{(k+1)} = \boldsymbol{z}^{(k)} - \beta^{(k)} a^{(k)} \nabla d(\boldsymbol{z}^{(k)} - \boldsymbol{x}_0) - \beta^{(k)} b^{(k)} \boldsymbol{s}(\boldsymbol{z}^{(k)}) \quad (6)$$

where $\beta^{(k)}$ is the step size within $[0, 1]$; $a^{(k)}$ and $b^{(k)}$ are two scalars, which will be specified later. As an intuitive explanation on Eq. (3), notice that the second term, which we will call the *distance reduction term*, reduces the distance to $\boldsymbol{x}_0$, whereas the third term, which we will call the *constraint reduction term*, reduces the the constraint (because $\boldsymbol{s}(\boldsymbol{z}^{(k)})$ and $\nabla c(\boldsymbol{z}^{(k)})$ has a positive inner product). Therefore, the projection move essentially strikes a balance between reduction in distance and reduction in constraint.

$a^{(k)}$ and $b^{(k)}$ can have two designs. The first design is to ensure the constraint values are roughly the same after the move, i.e. $c(\boldsymbol{z}^{(k)}) - c(\boldsymbol{x}^{(k+1)}) \approx 0$. By Taylor approximation, we have

$$\nabla^T c(\boldsymbol{z}^{(k)})(\boldsymbol{x}^{(k+1)} - \boldsymbol{z}^{(k)}) = 0, \quad (7)$$

whose solution is

$$b^{(k)} = \frac{a^{(k)} \nabla^T c(\boldsymbol{z}^{(k)}) \nabla d(\boldsymbol{z}^{(k)} - \boldsymbol{x}_0)}{\nabla^T c(\boldsymbol{z}^{(k)}) \boldsymbol{s}(\boldsymbol{z}^{(k)})}. \quad (8)$$

Another design is to ensure the perturbation norm reduces roughly by $\beta^{(k)}$, i.e. $d(\boldsymbol{x}^{(k+1)} - \boldsymbol{x}_0) \approx (1 - \beta^{(k)}) d(\boldsymbol{z}^{(k)} - \boldsymbol{x}_0)$. By Taylor approximation, we have

$$\nabla^T d(\boldsymbol{z}^{(k)} - \boldsymbol{x}_0)(\boldsymbol{x}^{(k+1)} - \boldsymbol{z}^{(k)}) = \beta^{(k)} d(\boldsymbol{z}^{(k)} - \boldsymbol{x}_0), \quad (9)$$

whose solution is

$$a^{(k)} = 1 - \frac{b^{(k)} \nabla^T d(\boldsymbol{z}^{(k)} - \boldsymbol{x}_0) \boldsymbol{s}(\boldsymbol{z}^{(k)})}{\nabla^T d(\boldsymbol{z}^{(k)} - \boldsymbol{x}_0) \nabla d(\boldsymbol{z}^{(k)} - \boldsymbol{x}_0)} \quad (10)$$

It should be noted that Eqs. (8) and (10) are just two specific choices for $a^{(k)}$ and $b^{(k)}$. It turns out that MARGINATTACK will work with a convergence guarantee for a wide range of bounded $a^{(k)}$s and $b^{(k)}$s that satisfy some conditions, as will be shown in section 3.4. Therefore, MARGINATTACK provides a general and flexible framework for zero-confidence adversarial attack designs. In practice, we find that Eq. (8) works better for $\ell_2$ norm, and Eq. (8) works better for $\ell_\infty$ norm.

---

[1]See Thm. 2 in the appendix for a detailed proof.

### 3.3 How MarginAttack Works

Figure 1 illustrates a typical convergence path of MARGINATTACK using $\ell_2$ norm and Eq. (8) as an example. The red dots on the right denote the original inputs $\boldsymbol{x}_0$ and its closest point on the decision boundary, $\boldsymbol{x}^*$. Suppose after iteration $k$, MARGINATTACK reaches $\boldsymbol{x}^{(k)}$, denoted by the green dot on the left. The restoration move travels directly towards the decision boundary by finding the normal direction to the current constraint contour. Then, the projection move travels along the tangent plane of the current constraint contour to reduce the distance to $\boldsymbol{x}_0$ while preventing the constraint value from deviating much. As intuitively expected, the iteration should eventually approach $\boldsymbol{x}^*$. Figure 2 plots an empirical convergence curve of the perturbation norm and constraint value of MARGINATTACK-$\ell_2$ on a randomly chosen CIFAR image. Each move from a triangle to a circle dot is a restoration move, and from circle to triangle a projection move. The red line is the smoothed version. As can be seen, a restoration move reduces the constraint value while slightly increasing the constraint norm, and a projection move reduces the perturbation norm while slightly affecting the constraint value. Both curves can eventually converge.

### 3.4 The Convergence Guarantee

The constraint function $c(\boldsymbol{x})$ in Eq. (2) is nonconvex, thus the convergence analysis for MARGINAT-TACK is limited to the vicinity of a unique local optimum, as stated in the following theorem.

**Theorem 1.** *Denote $\boldsymbol{x}^*$ as one local optimum for Eq. (1). Assume $\nabla c(\boldsymbol{x}^*)$ exists. Define projection matrices*

$$\boldsymbol{P} = \boldsymbol{I} - \boldsymbol{s}(\boldsymbol{x}^*)(\nabla^T c(\boldsymbol{x}^*)\boldsymbol{s}(\boldsymbol{x}^*))^{-1}\nabla^T c(\boldsymbol{x}^*) \tag{11}$$

*Consider the neighborhood $\mathbb{B} = \{\boldsymbol{x} : \|\boldsymbol{P}[\boldsymbol{x} - \boldsymbol{x}^*]\|_2^2 \leq X, |c(\boldsymbol{x})| \leq C\}$ that satisfies the following assumptions:*

1. *(Differentiability) $\forall \boldsymbol{x} \in \mathbb{B}$, $\nabla c(\boldsymbol{x})$ exists, but can be discontinuous, i.e. all the discontinuity points of the gradient in $\mathbb{B}$ are jump discontinuities;*

2. *(Lipschitz Continuity at $\boldsymbol{x}^*$) $\forall \boldsymbol{x} \in \mathbb{B}$, $\|\boldsymbol{s}(\boldsymbol{x}) - \boldsymbol{s}(\boldsymbol{x}^*)\|_2 \leq L_s\|\boldsymbol{s}(\boldsymbol{x}^*)\|_2\|\boldsymbol{x} - \boldsymbol{x}^*\|_2$;*

3. *(Bounded Gradient Norm) $\forall \boldsymbol{x} \in \mathbb{B}, 0 < m \leq \|\nabla c(\boldsymbol{x})\|_2 \leq M$;*

4. *(Bounded Gradient Difference) $\exists \delta > 0$, $\forall \boldsymbol{x}, \boldsymbol{y} \in \mathbb{B}$ s.t. $\boldsymbol{y} - \boldsymbol{x} = l\boldsymbol{s}(\boldsymbol{x})$ for some $l$,*

$$\nabla^T c(\boldsymbol{y})\boldsymbol{s}(\boldsymbol{x}) \geq \delta \nabla^T c(\boldsymbol{x})\boldsymbol{s}(\boldsymbol{x});$$

5. *(Constraint Convexity) $\exists \gamma \in (0, 1)$, $\forall \boldsymbol{x} \in \mathbb{B}$,*

$$(a^{(k)}\nabla d(\boldsymbol{x} - \boldsymbol{x}_0) + b^{(k)}\boldsymbol{s}(\boldsymbol{x}))^T \boldsymbol{P}^T \boldsymbol{P}(\boldsymbol{x} - \boldsymbol{x}_0) \geq \gamma(\boldsymbol{x} - \boldsymbol{x}_0)^T \boldsymbol{P}^T \boldsymbol{P}(\boldsymbol{x} - \boldsymbol{x}_0);$$

6. *(Unique Optimality) $\boldsymbol{x}^*$ is the only global optimum within $\mathbb{B}$;*

7. *(Constant Bounded Restoration Step Size) $\alpha^{(k)} = \alpha < M_\alpha$;[2]*

8. *(Shrinking Projected Step Size) $\beta^{(k)} < \beta/(k + k_0)^\nu$, where $0 < \nu < 1$ and $\beta \leq M_\beta$, $k_0 > m_k$;[3] $|a^{(k)}| < M_a$, $|b^{(k)}| < M_b$;*

9. *(Presence in Neighborhood) $\exists K$, $\boldsymbol{x}^{(K)} \in \text{int}[\mathbb{B}]$, i.e. the interior of $\mathbb{B}$.*

*Then we have the convergence guarantee $\lim_{k \to \infty}\|\boldsymbol{x}^{(k)} - \boldsymbol{x}^*\|_2 = 0$.*

The proof will be presented in the appendix. Here are a few remarks. First, assumption 1 allows jump discontinuities in $\nabla c(\boldsymbol{x})$ almost everywhere, which is a very practical assumption for deep neural networks. Most neural network operations, such as ReLU and max-pooling, as well as the max operation in Eq. (2), introduce nothing beyond jump discontinuities in gradient.

---

[2]See Eq. (19) for the definition of $M_\alpha$ in the appendix.
[3]See Eqs. (50) and (20) for the definitions of $M_\beta$ and $m_k$ in the appendix.

Second, assumption 3 does require the constraint gradient to be lower bounded, which may lead to concerns that MARGINATTACK may fail in the presence of gradient masking (Papernot et al., 2017). However, notice that the gradient boundedness assumption is only imposed in $\mathbb{B}$, which is in the vicinity of the decision boundary, whereas gradient masking is most likely to appear away from the decision boundary and where the input features are populated. Besides, as will be discussed later, a random initialization as in PGD will be adopted to bypass regions with gradient masking. Experiments on adversarially trained models also verify the robustness of MARGINATTACK.

Finally, assumption 5 essentially stipulates that $c(\boldsymbol{x})$ is convex or "not too concave" in $\mathbb{B}$ (and thus so is the constraint set $c(\boldsymbol{x}) \leq 0$), so that the first order optimality condition can readily imply local minimum instead of a local maximum. In fact, it can be shown that assumption 5 can be implied if $c(\boldsymbol{x})$ is convex in $\mathbb{B}$.[4]

### 3.5 Additional Implementation Details

There are a few additional implementation details as outlined below.

**Box Constraint:** In many applications, each dimension of the input features should be bounded, i.e. $\boldsymbol{x} \in [x_{\min}, x_{\max}]^n$. To impose the box constraint, the restoration move problem as in Eq. (3) is modified as

$$\boldsymbol{z}^{(k)} = \underset{\boldsymbol{x} \in [x_{\min}, x_{\max}]^n}{\arg\min} \ d(\boldsymbol{x} - \boldsymbol{x}^{(k)}), \quad \text{s.t.} \ \nabla^T c(\boldsymbol{x}^{(k)})(\boldsymbol{x} - \boldsymbol{x}^{(k)}) = -\alpha^{(k)} c(\boldsymbol{x}^{(k)}), \tag{12}$$

whose solution is

$$\boldsymbol{z}^{(k)} = \text{Proj}_{[x_{\min}, x_{\max}]^n}\{\tilde{\boldsymbol{z}}^{(k)}\}, \quad \text{where} \ \tilde{\boldsymbol{z}}^{(k)} = \boldsymbol{x}^{(k)} - \frac{\alpha^{(k)} c(\boldsymbol{x}^{(k)}) + \sum_{i \in \mathbb{I}^C} \nabla_i c(\boldsymbol{x}^{(k)})(\boldsymbol{z}_i^{(k)} - \boldsymbol{x}_i^{(k)})}{\sum_{i \in \mathbb{I}} \nabla_i c(\boldsymbol{x}^{(k)}) \boldsymbol{s}_i(\boldsymbol{x}^{(k)})} \boldsymbol{s}(\boldsymbol{x}^{(k)}). \tag{13}$$

$\text{Proj}(\cdot)$ is an operator that projects the vector in its argument onto the subset in its subscript. $\mathbb{I}$ is a set of indices with which the elements in $\tilde{\boldsymbol{z}}^{(k)}$ satisfy the box constraint, and $\mathbb{I}^C$ is its complement. $\mathbb{I}$ is determined by running Eq. (13) iteratively and updating $\mathbb{I}$ after each iterations.

Unlike other attack algorithms that simply project the solution onto the constraint box, MARGINATTACK incorporates the box constraint in a principled way, such that any local optimal solution $\boldsymbol{x}^*$ will be an invariant point of the restoration move. Thus the convergence is faster.

**Target Scan:** According to Eq. (2), each restoration move essentially approaches the adversarial class with the highest logit, but the class with the highest logit may not be the closest. To mitigate the problem, we follow a similar approach adopted in DeepFool, which we call *target scan*. Target scan performs a target-specific restoration move towards each class, and chooses the move with the shortest distance. Formally, target scan introduces a set of target-specific constraints $\{c_i(\boldsymbol{x}) = l_t(\boldsymbol{x}) - l_i(\boldsymbol{x}) - \varepsilon\}$. A restoration move with target scan solves

$$\boldsymbol{z}^{(k)} = \underset{i \in \mathbb{A}}{\arg\min} \ d(\boldsymbol{z}^{(k,i)} - \boldsymbol{x}_0) \tag{14}$$

where $\boldsymbol{z}^{(k,i)}$ is the solution to Eqs. (3) or (12) with $c(\boldsymbol{x}^{(k)})$ replaced with $c_i(\boldsymbol{x}^{(k)})$, and thus is equal to Eqs. (4) or (13) with $c(\boldsymbol{x}^{(k)})$ replaced with $c_i(\boldsymbol{x}^{(k)})$. $\mathbb{A}$ is a set of candidate adversarial calsses, which can be all the incorrect classes if the number of classes is small, or which can be a subset of the adversarial classes with the highest logits otherwise. Experiments show that target scan is necessary only in the first few restoration moves, when the closest and highest adversarial classes are likely to be distinct. Therefore, the computation cost will not increase too much.

**Initialization:** The initialization of $\boldsymbol{x}^{(0)}$ can be either deterministic or random as follows

$$\boldsymbol{x}^{(0)} = \boldsymbol{x}_0 \ \text{(Deterministic)}, \qquad \boldsymbol{x}^{(0)} = \boldsymbol{x}_0 + \boldsymbol{u}, \ \boldsymbol{u} \sim \mathcal{U}\{[-u, u]^n\} \ \text{(Random)} \tag{15}$$

where $\mathcal{U}\{[-u, u]^n\}$ denotes the uniform random distribution in $[-u, u]^n$. Similar to PGD, we can perform multiple trials with random initialization to find a better local optimum.

**Final Tuning** MARGINATTACK can only cause misclassification when $c(\boldsymbol{x}) \leq \varepsilon$. To make sure the attack is successful, the final iterations of MARGINATTACK consists of restoration moves only,

---

[4]See Thm. 3 in the appendix.

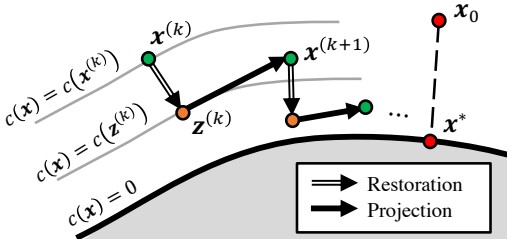

Figure 1: A convergence path of MARGINATTACK.

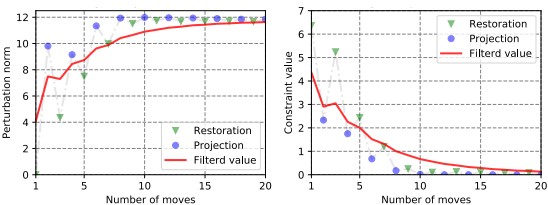

Figure 2: An empirical convergence curve of perturbation norm (left) and constraint value (right).

**Algorithm 1:** MARGINATTACK Procedure

**Input** : A set of logit functions $l_{0:C-1}(\boldsymbol{x})$;
an input feature $\boldsymbol{x}_0$ and its label $t$;
**Output:** A solution $\tilde{\boldsymbol{x}}^*$ to Eq. (1)
Initialize $\boldsymbol{x}^{(0)}$ according to Eq. (15);
**for** $k <$ *number of iterations* **do**
    **if** $k <$ *number of target scan iterations* **then**
        Do target scan restoration move as in
        Eq. (14);
    **else**
        Do regular restoration move as in
        Eqs. (3) or (12);
    **end**
    **if** $k <$ *final tuning iteration* **then**
        Do projection move as in Eqs. (6);
    **else**
        Skip projection move: $\boldsymbol{x}^{(k+1)} = \boldsymbol{z}^{(k)}$;
    **end**
**end**
$\tilde{\boldsymbol{x}}^* = \boldsymbol{x}^{(k)}$.

and no projection moves, until a misclassification is caused. This can also ensure the final solution satisfies the box constraint (because only the restoration move incorporates the box constraint).

**Summary:** Alg. 1 summarizes the MARGINATTACK procedure. As for the complexity, each restoration move or projection move requires only one backward propagation, and thus the computational complexity of each move is comparable to one iteration of most attack algorithms.

## 4 EXPERIMENTS

This section compares MARGINATTACK with several state-of-the-art adversarial attack algorithms in terms of the perturbation norm and computation time on image classification benchmarks.

### 4.1 ATTACKING REGULAR MODELS

#### 4.1.1 CONFIGURATIONS

Three regularly trained models are evaluated on.

- **MNIST** (LeCun et al., 1998)**:** The classifier is a stack of two $5 \times 5$ convolutional layers with 32 and 64 filters respectively, followed by two fully-connected layers with 1,024 hidden units.
- **CIFAR10** (Krizhevsky & Hinton, 2009)**:** The classifier is a pre-trained ResNet32 (He et al., 2016) provided by TensorFlow.[5].
- **ImageNet** (Russakovsky et al., 2015)**:** The classifier is a pre-trained ResNet50 (He et al., 2016) provided by TensorFlow Keras[6]. Evaluation is on a validation subset containing 10,000 images.

The range of each pixel is $[0, 1]$ for MNIST, and $[0, 255]$ for CIFAR10 and ImageNet. The settings of MARGINATTACK and baselines are listed below. Unless stated otherwise, the baseline algorithms are implemented by cleverhans (Nicolas Papernot, 2017). The hyperparameters are set to defaults if not specifically stated.

- **CW** (Carlini & Wagner, 2017)**:** The target and evaluation norm is $\ell_2$. The learning rate is set to 0.05 for MNIST, 0.001 for CIFAR10 and 0.01 for ImageNet, which are tuned to its best performance. The number of binary steps for multiplier search is 10.

---
[5]https://github.com/tensorflow/models/tree/master/official
[6]https://www.tensorflow.org/api_docs/python/tf/keras/applications/ResNet50

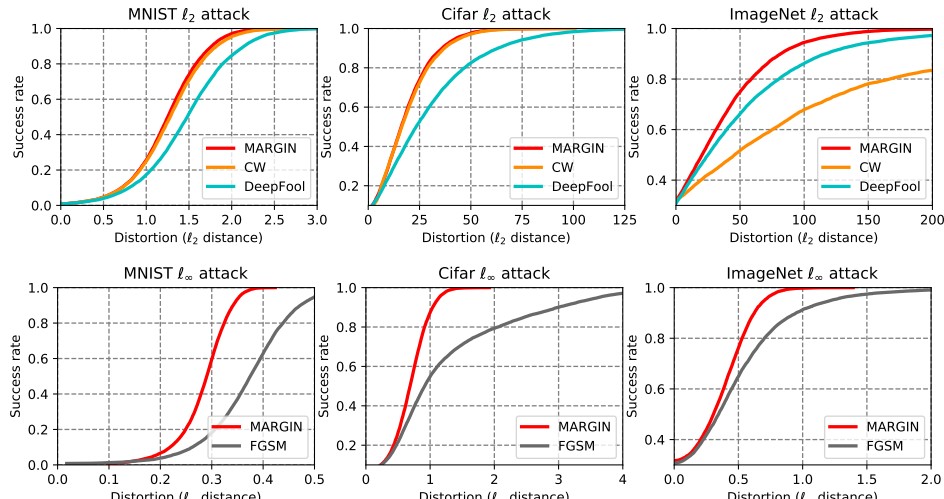

Figure 3: Adversarial attacks on (left) MNIST, (middle) Cifar, and (right) ImageNet dataset.

- **DeepFool** (Moosavi Dezfooli et al., 2016)**:** The evaluation norm is $\ell_2$.
- **FGSM** (Goodfellow et al., 2014)**:** FGSM is implemented by authors. The step size is searched to achieve zero-confidence attack. The evaluation distance metric is $\ell_\infty$.
- **PGD** (Madry et al., 2017)**:** The target and evaluation norm are $\ell_\infty$. The learning rate is set to 0.01 for MNIST, and 0.05 for CIFAR10 and 0.1 for ImageNet.
- **MARGINATTACK:** Two versions of MARGINATTACK are implemented, whose target and evaluation norms are $\ell_2$, and $\ell_\infty$, respectively. The hyperparmeters are detailed in Table 4 in the appendix. The first 10 restoration moves are with target scan, and the last 20 moves are all restoration moves.

The number of iterations/moves is set to 2,000 for CW, 200 with 10 random starts for PGD and MARGINATTACK (except for ImageNet where there is only one random run), and 200 for the rest.

### 4.1.2 RESULTS AND ANALYSES

Except for PGD, all the other attacks are zero-confidence attacks. For these attacks, we plot the CDF of the margins of the validation data, which can also be interpreted as the percentage success rate of these attacks as a function of perturbation level. Figure 3 plots the success rate curves, where the upper panel shows the $\ell_2$ attacks, and the lower one shows $\ell_\infty$ attacks. As can be observed, the MARGINATTACK curves are above all other algorithms at all perturbation levels and in all datasets. CW is very close to MARGINATTACK on MNIST and CIFAR10, but MARGINATTACK maintains a 3% advantage on MNIST and 1% on CIFAR10. It seems that CW is unable to converge well within 2,000 iterations on ImageNet, although the learning rate has been tuned to maximize its performance. MARGINATTACK, on the other hand, converges more efficiently and consistently.

To obtain a success rate curve for PGD, we have to run the attack again and again for many different perturbation levels, which can be time-consuming for large datasets (this shows an advantage of zero-confidence attacks over fix-perturbation attacks). Instead, we choose four perturbation levels for each attack scenario to compare. The perturbation levels are chosen to roughly follow the 0.2, 0.4, 0.6 and 0.8 quantiles of the MARGINATTACK margins. Table 1 compares the success rates under the chosen quantiles among the $\ell_\infty$ attacks. We can see that MARGINATTACK outperforms PGD under all the perturbation levels, and that both significantly dominate FGSM.

### 4.2 ATTACKING ADVERSARIALLY TRAINED MODEL

We also evaluate MARGINATTACK on the MNIST Adversarial Examples Challenge[7], which is a challenge of attacking an MNIST model adversarially trained using PGD with 0.3 perturbation level.

---

[7]https://github.com/MadryLab/mnist_challenge

Table 1: Success rate (%) of adversarial attacks under given perturbation norms.

| Algorithm | MNIST | CIFAR | IMAGENET |
|---|---|---|---|
| | 0.06 / 0.08 / 0.10 / 0.12 | 0.2 / 0.4 / 0.6 / 1 | 0.05 / 0.1 / 0.2 / 0.4 |
| FGSM | 7.55 / 13.9 / 24.9 / 35.4 | 18.5 / 31.0 / 41.1 / 54.7 | 39.8 / 47.2 / 60.1 / 75.3 |
| PGD | 17.1 / 42.2 / 73.7 / 91.8 | 18.9 / 38.9 / 59.1 / 84.1 | 40.4 / 49.8 / 68.8 / 90.6 |
| Ours | 18.1 / 43.0 / 74.1 / 92.1 | 21.1 / 42.2 / 62.6 / 87.3 | 41.5 / 51.3 / 69.0 / 90.8 |

Table 2: Success rate under 0.3 perturbation norm of the MNIST Adversarial Examples Challenge.

| Algorithm | Success Rate (%) |
|---|---|
| Zheng et al. (2018) | 11.21 |
| MARGINATTACK ($\ell_\infty$) | 11.16 |
| 1st-Order on Logit Diff | 11.15 |
| PGD on Cross-Entropy Loss | 10.38 |
| PGD on CW Loss | 10.29 |

Table 3: Running time comparison (in seconds) on a single batch of images.

| Algorithm | MNIST | CIFAR | IMAGENET |
|---|---|---|---|
| CW | 16.02 | 234.75 | 872.28 |
| DeepFool | 1.14 | 21.26 | 44.41 |
| PGD | 0.87 | 33.17 | 46.3 |
| FSGM | 0.11 | 0.95 | 10.05 |
| Ours ($\ell_2$) | 3.01 | 51.03 | 248.82 |

Same as the PGD baseline listed, MARGINATTACK is run with 50 random starts, and the initialization perturbation range $u = 0.3$. The number of moves is 500. The target norm is $\ell_\infty$. $b_n = 5$ and $a_n$ is set as in Eq. (10). The rest of the configuration is the same as in the previous experiments.

Table 2 lists the success rates of different attacks under 0.3 perturbation level. The baseline algorithms are all fix-perturbation attacks, and their results are excerpted from the challenge white-box attack leaderboard. As can be seen, MARGINATTACK, as the only zero-confidence attack algorithm, has the second best result, which shows that it performs competitively against the state-of-the-art fix-perturbation attacks.

### 4.3 CONVERGENCE

We would like to revisit the convergence plot of the constraint value $c(\boldsymbol{x})$ and perturbation norm $d(\boldsymbol{x})$ of as in Fig. 2. We can see that MARGINATTACK converges very quickly. In the example shown in the figure, it is able to converge within 20 moves. Therefore, MARGINATTACK can be greatly accelerated. If margin accuracy is the priority, a large number of moves, *e.g.* 200 as in our experiment, would help. However, if efficiency is the priory, a small number of moves, *e.g.* 30, suffices to produce a decent attack.

To further assess the efficiency of MARGINATTACK, Tab. 3 compares the running time (in seconds) of attacking one batch of images, implemented on a single NVIDIA TESLA P100 GPU. The batch size is 200 for MNIST and CIFAR10, and 100 for ImageNet. The settings are the same as stated in section 4.1, except that for a better comparison, the number of iterations of CW is cut down to 200, and PGD and MARGINATTACK runs one random pass, so that all the algorithms have the same iteration/moves. Only the $\ell_2$ versions of MARGINATTACK are shown because the other versions have similar run times. As shown, running time of MARGINATTACK is much shorter than CW, and is comparable to DeepFool and PGD. CW is significantly slower that the other algorithms because it has to run multiple trials to search for the best Lagrange multiplier. Note that DeepFool and CW enable early stop, but MARGINATTACK does not. Considering MARGINATTACK's fast convergence rate, the running time can be further reduced by early stop.

### 5 CONCLUSION

We have proposed MARGINATTACK, a novel zero-confidence adversarial attack algorithm that is better able to find a smaller perturbation that results in misclassification. Both theoretical and empirical analyses have demonstrated that MARGINATTACK is an efficient, reliable and accurate adversarial attack algorithm, and establishes a new state-of-the-art among zero-confidence attacks. What is more, MARGINATTACK still has room for improvement. So far, only two settings of $a^{(k)}$ and $b^{(k)}$ are developed, but MARGINATTACK will work for many other settings, as long as assumption 5 is satisfied. Authors hereby encourage exploring novel and better settings for the MARGINATTACK framework, and promote MARGINATTACK as a new robustness evaluation measure or baseline in the field of adversarial attack and defense.

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

APPENDIX

A   PROVING THM. 1

This supplementary material aims to prove Thm. 1. Without the loss of generality, $K$ in Eq. (9) in set to 0. Before we prove the theorem, we need to introduce some lemmas.

**Lemma 1.1.** *If assumption 3 in Thm. 1 holds, then $\forall \boldsymbol{x} \in \mathbb{B}$*

$$\frac{\nabla^T c(\boldsymbol{x}) \boldsymbol{s}(\boldsymbol{x})}{\|\boldsymbol{s}(\boldsymbol{x})\|_2} \geq \frac{m}{\sqrt{n}} \tag{16}$$

*Proof.* According to Eq. (5), for $\ell_2$ norm,

$$\frac{\nabla^T c(\boldsymbol{x}) \boldsymbol{s}(\boldsymbol{x})}{\|\boldsymbol{s}(\boldsymbol{x})\|_2} = \|\nabla c(\boldsymbol{x})\|_2 \geq m > \frac{m}{\sqrt{n}} \tag{17}$$

for $\ell_\infty$ norm,

$$\frac{\nabla^T c(\boldsymbol{x}) \boldsymbol{s}(\boldsymbol{x})}{\|\boldsymbol{s}(\boldsymbol{x})\|_2} = \frac{\|\nabla c(\boldsymbol{x})\|_1}{\|\boldsymbol{s}(\boldsymbol{x})\|_2} \geq \frac{\|\nabla c(\boldsymbol{x})\|_2}{\sqrt{n}} \geq \frac{m}{\sqrt{n}} \tag{18}$$

$\square$

**Lemma 1.2.** *Given all the assumptions in Thm. 1, where*

$$M_\alpha = \min \left\{ 1, \sup_{\substack{\boldsymbol{x}, \boldsymbol{y} \in \mathbb{B}: \\ \exists l, \boldsymbol{y} - \boldsymbol{x} = l \boldsymbol{s}(\boldsymbol{x})}} \frac{\nabla^T c(\boldsymbol{y}) \boldsymbol{s}(\boldsymbol{x})}{\nabla^T c(\boldsymbol{x}) \boldsymbol{s}(\boldsymbol{x})}, \frac{m\gamma}{2nL_s \kappa} \right\} \tag{19}$$

$$m_k = \left( A^{-1/2\nu} - 1 \right)^{-1} \tag{20}$$

*and assuming $\boldsymbol{x}^{(k)}, \boldsymbol{z}^{(k)} \in \mathbb{B}, \forall k$, then we have*

$$|c(\boldsymbol{x}^{(k)})| \leq \kappa \beta^{(k)} \tag{21}$$

*where*

$$\kappa = \max \left\{ \frac{B}{\sqrt{A}(1 - \sqrt{A})}, \frac{c(\boldsymbol{x}^{(0)})}{\beta^{(0)}} \right\} \tag{22}$$

*$A$ and $B$ are defined in Eq. (32).*

*According to assumption 8, this implies*

$$\lim_{k \to \infty} |c(\boldsymbol{x}^{(k)})| = 0 \tag{23}$$

*at the rate of at least $1/n^\nu$.*

*Proof.* As a digression, the second term in Eq. (19) is well defined, because

$$\frac{\nabla^T c(\boldsymbol{y}) \boldsymbol{s}(\boldsymbol{x})}{\nabla^T c(\boldsymbol{x}) \boldsymbol{s}(\boldsymbol{x})}$$

is upper bounded by Lem. 1.1 and assumptions 3.

Back to proving the lemma, we will prove that each restoration move will bring $c(\boldsymbol{x}^{(k)})$ closer to 0, while each projection move will not change $c(\boldsymbol{x}^{(k)})$ much.

First, for the restoration move

$$
\begin{aligned}
|c(\boldsymbol{z}^{(k)})| &\leq |c(\boldsymbol{x}^{(k)}) + \nabla^T c(\boldsymbol{\xi})(\boldsymbol{z}^{(k)} - \boldsymbol{x}^{(k)})| \\
&= \left| c(\boldsymbol{x}^{(k)}) - \alpha^{(k)} \nabla^T c(\boldsymbol{\xi}) \frac{c(\boldsymbol{x}^{(k)}) \boldsymbol{s}(\boldsymbol{x}^{(k)})}{\nabla^T c(\boldsymbol{x}^{(k)}) \boldsymbol{s}(\boldsymbol{x}^{(k)})} \right| \\
&= \left| 1 - \alpha \frac{\nabla^T c(\boldsymbol{\xi}) \boldsymbol{s}(\boldsymbol{x}^{(k)})}{\nabla^T c(\boldsymbol{x}^{(k)}) \boldsymbol{s}(\boldsymbol{x}^{(k)})} \right| |c(\boldsymbol{x}^{(k)})| \\
&\leq (1 - \alpha\delta)|c(\boldsymbol{x}^{(k)})|
\end{aligned}
\tag{24}
$$

The first line is from the generalization of Mean-Value Theorem with jump discontinuities, and $\boldsymbol{\xi} = t\boldsymbol{z}^{(k)} + (1-t)\boldsymbol{x}^{(k)}$ and $t$ is a real number in $[0, 1]$. The second line is from Eq. (4). The last line is from assumptions 4 and 7 and Eq. (19).

Next, for the projection move

$$
\begin{aligned}
|c(\boldsymbol{x}^{(k+1)})| &\leq |c(\boldsymbol{z}^{(k)})| + M\|\boldsymbol{x}^{(k+1)} - \boldsymbol{z}^{(k)}\|_2 \\
&= |c(\boldsymbol{z}^{(k)})| + \beta^{(k)} M \|a^{(k)} \nabla d(\boldsymbol{z}^{(k)} - \boldsymbol{x}_0) + b^{(k)} \boldsymbol{s}(\boldsymbol{z}^{(k)})\|_2 \\
&\leq |c(\boldsymbol{z}^{(k)})| + \beta^{(k)} M \left( a^{(k)} \|\nabla d(\boldsymbol{z}^{(k)} - \boldsymbol{x}_0)\|_2 + b^{(k)} \|\boldsymbol{s}(\boldsymbol{z}^{(k)})\|_2 \right)
\end{aligned}
\tag{25}
$$

The first line is from the fact that assumption 3 implies that $c(\boldsymbol{x})$ is $M$-Lipschitz continuous.

Both $\|\nabla d(\boldsymbol{z}^{(k)} - \boldsymbol{x}_0)\|_2$ and $\|\boldsymbol{s}(\boldsymbol{z}^{(k)})\|_2$ is upper-bounded, i.e.

$$
\|\nabla d(\boldsymbol{z}^{(k)} - \boldsymbol{x}_0)\|_2 < M_d, \qquad \|\boldsymbol{s}(\boldsymbol{z}^{(k)})\|_2 < M_s
\tag{26}
$$

for some $M_d$ and $M_s$. To see this, for $\ell_2$ norm

$$
\|\nabla d(\boldsymbol{z}^{(k)} - \boldsymbol{x}_0)\|_2 = 2\|\boldsymbol{z}^{(k)} - \boldsymbol{x}_0\|_2 \leq 2b, \qquad \|\boldsymbol{s}(\boldsymbol{z}^{(k)})\|_2 = 1
\tag{27}
$$

where $b$ is defined as the maximum perturbation norm ($\ell_2$) within $\mathbb{B}$, *i.e.*

$$
b = \max_{x \in \mathbb{B}} \|\boldsymbol{x}_0 - \boldsymbol{x}\|_2
\tag{28}
$$

which is well defined because $\mathbb{B}$ is a tight set. For $\ell_\infty$ norm,

$$
\|\nabla d(\boldsymbol{z}^{(k)} - \boldsymbol{x}_0)\|_2 \leq \sqrt{n}, \qquad \|\boldsymbol{s}(\boldsymbol{z}^{(k)})\|_2 \leq \sqrt{n}
\tag{29}
$$

Note that Eq. (26) also holds for other norms. With Eq. (26) and assumption 8, Eq. (25) becomes

$$
|c(\boldsymbol{x}^{(k+1)})| \leq |c(\boldsymbol{z}^{(k)})| + \beta^{(k)} M (M_a M_d + M_b M_s)
\tag{30}
$$

Combining Eqs. (24) and (30) we have

$$
|c(\boldsymbol{x}^{(k+1)})| \leq A|c(\boldsymbol{x}^{(k)})| + \beta^{(k)} B
\tag{31}
$$

where

$$
\begin{aligned}
A &= 1 - \alpha\delta \\
B &= M (M_a M_d + M_b M_s)
\end{aligned}
\tag{32}
$$

According to assumption 7, $0 < A < 1$. Also, according to Eq. (20), $\beta^{(k)}/\beta^{(k+1)} \leq A^{-1/2}, \forall k$,

Divide Eq (31) by $\beta^{(k)}$, we have

$$
\frac{|c(\boldsymbol{x}^{(k+1)})|}{\beta^{(k+1)}} \leq \frac{|c(\boldsymbol{x}^{(k+1)})|}{\sqrt{A}\beta^{(k)}} \leq \sqrt{A}\frac{|c(\boldsymbol{x}^{(k)})|}{\beta^{(k)}} + \frac{B}{\sqrt{A}}
\tag{33}
$$

and thus

$$
\frac{|c(\boldsymbol{x}^{(k+1)})|}{\beta^{(k+1)}} - \frac{B}{\sqrt{A}(1 - \sqrt{A})} \leq \sqrt{A} \left( \frac{|c(\boldsymbol{x}^{(k)})|}{\beta^{(k)}} - \frac{B}{\sqrt{A}(1 - \sqrt{A})} \right)
\tag{34}
$$

If

$$
\frac{|c(\boldsymbol{x}^{(0)})|}{\beta^{(0)}} - \frac{B}{\sqrt{A}(1 - \sqrt{A})} \leq 0
$$

Then Eq. (34) implies

$$
\frac{|c(\boldsymbol{x}^{(k)})|}{\beta^{(k)}} - \frac{B}{\sqrt{A}(1 - \sqrt{A})} \leq 0, \forall k
\tag{35}
$$

Otherwise, Eq. (34) implies

$$
\frac{|c(\boldsymbol{x}^{(k)})|}{\beta^{(k)}} - \frac{B}{\sqrt{A}(1 - \sqrt{A})} \leq \frac{|c(\boldsymbol{x}^{(0)})|}{\beta^{(0)}} - \frac{B}{\sqrt{A}(1 - \sqrt{A})}, \forall k
\tag{36}
$$

This concludes the proof. $\qquad \square$

**Lemma 1.3.** *Given all the assumptions in Thm. 1, and assuming $\boldsymbol{x}^{(k)}, \boldsymbol{z}^{(k)} \in \mathbb{B}, \forall k$, we have*

$$\lim_{k \to \infty} \|\boldsymbol{P}(\boldsymbol{x}^{(k)} - \boldsymbol{x}_0)\|_2^2 = 0 \tag{37}$$

*Proof.* First, for restoration move

$$\|\boldsymbol{P}[\boldsymbol{z}^{(k)} - \boldsymbol{x}_0]\|_2^2$$
$$= \|\boldsymbol{P}[\boldsymbol{x}^{(k)} - \boldsymbol{x}_0]\|_2^2 + 2[\boldsymbol{z}^{(k)} - \boldsymbol{x}_0]^T \boldsymbol{P}^T \boldsymbol{P}[\boldsymbol{z}^{(k)} - \boldsymbol{x}^{(k)}] - \|\boldsymbol{P}[\boldsymbol{z}^{(k)} - \boldsymbol{x}^{(k)}]\|_2^2$$
$$\leq \|\boldsymbol{P}[\boldsymbol{x}^{(k)} - \boldsymbol{x}_0]\|_2^2 + 2\|\boldsymbol{P}[\boldsymbol{z}^{(k)} - \boldsymbol{x}_0]\|_2 \|\boldsymbol{P}[\boldsymbol{z}^{(k)} - \boldsymbol{x}^{(k)}]\|_2$$
$$= \|\boldsymbol{P}[\boldsymbol{x}^{(k)} - \boldsymbol{x}_0]\|_2^2 + 2\alpha\|\boldsymbol{P}[\boldsymbol{z}^{(k)} - \boldsymbol{x}_0]\|_2 \left\| \frac{c(\boldsymbol{x}^{(k)})}{\nabla^T c(\boldsymbol{x}^{(k)}) s(\boldsymbol{x}^{(k)})} \boldsymbol{P} s(\boldsymbol{x}^{(k)}) \right\|_2$$
$$= \|\boldsymbol{P}[\boldsymbol{x}^{(k)} - \boldsymbol{x}_0]\|_2^2 + 2\alpha\|\boldsymbol{P}[\boldsymbol{z}^{(k)} - \boldsymbol{x}_0]\|_2 \left\| \frac{c(\boldsymbol{x}^{(k)})}{\nabla^T c(\boldsymbol{x}^{(k)}) s(\boldsymbol{x}^{(k)})} \boldsymbol{P}[s(\boldsymbol{x}^{(k)}) - s(\boldsymbol{x}^*)] \right\|_2$$
$$= \|\boldsymbol{P}[\boldsymbol{x}^{(k)} - \boldsymbol{x}_0]\|_2^2 + 2\alpha\|\boldsymbol{P}[\boldsymbol{z}^{(k)} - \boldsymbol{x}_0]\|_2 \left\| \frac{c(\boldsymbol{x}^{(k)}) \nabla^T c(\boldsymbol{x}^*)(s(\boldsymbol{x}^{(k)}) - s(\boldsymbol{x}^*))}{\nabla^T c(\boldsymbol{x}^{(k)}) s(\boldsymbol{x}^{(k)}) \nabla^T c(\boldsymbol{x}^*) s(\boldsymbol{x}^*)} s(\boldsymbol{x}^*) \right\|_2$$
$$\leq \|\boldsymbol{P}[\boldsymbol{x}^{(k)} - \boldsymbol{x}_0]\|_2^2 + \frac{2\alpha n |c(\boldsymbol{x}^{(k)})|}{m} \|\boldsymbol{P}[\boldsymbol{z}^{(k)} - \boldsymbol{x}_0]\|_2 \frac{\|s(\boldsymbol{x}^{(k)}) - s(\boldsymbol{x}^*)\|_2}{\|s(\boldsymbol{x}^{(k)})\|_2}$$
$$\leq \|\boldsymbol{P}[\boldsymbol{x}^{(k)} - \boldsymbol{x}_0]\|_2^2 + \frac{2\alpha n L_s \kappa \beta^{(k)}}{m} \|\boldsymbol{P}[\boldsymbol{z}^{(k)} - \boldsymbol{x}_0]\|_2 \|\boldsymbol{x}^{(k)} - \boldsymbol{x}^*\|_2$$
$$\leq \|\boldsymbol{P}[\boldsymbol{x}^{(k)} - \boldsymbol{x}_0]\|_2^2 + \frac{2\alpha n L_s \kappa \beta^{(k)}}{m} \|\boldsymbol{P}[\boldsymbol{z}^{(k)} - \boldsymbol{x}_0]\|_2 \left[ \|\boldsymbol{P}[\boldsymbol{x}^{(k)} - \boldsymbol{x}_0]\|_2 + \frac{c(\boldsymbol{x}^{(k)}) + M\|\boldsymbol{P}[\boldsymbol{x}^{(k)} - \boldsymbol{x}_0]\|_2}{\delta m / \sqrt{n}} \right]$$
$$\leq \|\boldsymbol{P}[\boldsymbol{x}^{(k)} - \boldsymbol{x}_0]\|_2^2 + \frac{2\alpha n L_s \kappa \beta^{(k)}(m + M\sqrt{n})}{\delta m^2} \|\boldsymbol{P}[\boldsymbol{z}^{(k)} - \boldsymbol{x}_0]\|_2 \|\boldsymbol{P}[\boldsymbol{x}^{(k)} - \boldsymbol{x}_0]\|_2 + \frac{2\alpha \sqrt{n} L_s \kappa^2 \beta^{(k)2}}{\delta m^2}$$
$$\tag{38}$$

Line 4 is given by Eq. (3). Line 5 is derived from Lem. 1.1. The last line is from Lem. 1.2.

Eq. (38) implies

$$(\|\boldsymbol{P}[\boldsymbol{z}^{(k)} - \boldsymbol{x}_0]\|_2 - r_1^{(k)} \|\boldsymbol{P}[\boldsymbol{x}^{(k)} - \boldsymbol{x}_0]\|_2)(\|\boldsymbol{P}[\boldsymbol{z}^{(k)} - \boldsymbol{x}_0]\|_2 - r_2^{(k)} \|\boldsymbol{P}[\boldsymbol{x}^{(k)} - \boldsymbol{x}_0]\|_2) \leq \frac{2\alpha \sqrt{n} L_s \kappa^2 \beta^{(k)2}}{\delta m^2} \tag{39}$$

where

$$r_1^{(k)} = \frac{1}{2} \left( \frac{2\alpha n L_s \kappa \beta^{(k)}}{m} - \sqrt{\left( \frac{2\alpha n L_s \kappa \beta^{(k)}}{m} \right)^2 + 4} \right) < 0$$
$$r_2^{(k)} = \frac{1}{2} \left( \frac{2\alpha n L_s \kappa \beta^{(k)}}{m} + \sqrt{\left( \frac{2\alpha n L_s \kappa \beta^{(k)}}{m} \right)^2 + 4} \right) > 0 \tag{40}$$

It can easily be shown that

$$\frac{-r_1^{(k)}}{r_2^{(k)}} = \frac{r_1^{(k)2}}{-r_1^{(k)} r_2^{(k)}} = \frac{r_1^{(k)2}}{4} \geq \frac{r_1^{(0)2}}{4} \tag{41}$$

Therefore

$$\|\boldsymbol{P}[\boldsymbol{z}^{(k)} - \boldsymbol{x}_0]\|_2 - r_1^{(k)} \|\boldsymbol{P}[\boldsymbol{x}^{(k)} - \boldsymbol{x}_0]\|_2 \geq \frac{r_1^{(0)2}}{4} (\|\boldsymbol{P}[\boldsymbol{z}^{(k)} - \boldsymbol{x}_0]\|_2 + r_2^{(k)} \|\boldsymbol{P}[\boldsymbol{x}^{(k)} - \boldsymbol{x}_0]\|_2) \tag{42}$$

Combining Eqs. (39) and (42), we have

$$\|\boldsymbol{P}[\boldsymbol{z}^{(k)} - \boldsymbol{x}_0]\|_2^2 \leq r_2^{(k)2} \|\boldsymbol{P}[\boldsymbol{x}^{(k)} - \boldsymbol{x}_0]\|_2^2 + \frac{8\alpha \sqrt{n} L_s \kappa^2 \beta^{(k)2}}{\delta m^2 r_1^{(0)2}}$$
$$< \left( 1 + \frac{2\alpha n L_s \kappa \beta^{(k)}}{m} \right)^2 \|\boldsymbol{P}[\boldsymbol{x}^{(k)} - \boldsymbol{x}_0]\|_2^2 + \frac{8\alpha \sqrt{n} L_s \kappa^2 \beta^{(k)2}}{\delta m^2 r_1^{(0)2}} \tag{43}$$

Next, for projection move

$$\|\boldsymbol{P}[\boldsymbol{x}^{(k+1)} - \boldsymbol{x}_0]\|_2^2$$
$$= \|\boldsymbol{P}[\boldsymbol{z}^{(k)} - \boldsymbol{x}_0]\|_2^2 + 2[\boldsymbol{x}^{(k+1)} - \boldsymbol{z}^{(k)}]^T \boldsymbol{P}[\boldsymbol{z}^{(k)} - \boldsymbol{x}_0] + \|\boldsymbol{x}^{(k+1)} - \boldsymbol{z}^{(k)}\|_2^2$$
$$= \|\boldsymbol{P}[\boldsymbol{z}^{(k)} - \boldsymbol{x}_0]\|_2^2 + 2\beta^{(k)}[a^{(k)}\nabla d(\boldsymbol{z}^{(k)} - \boldsymbol{x}_0) + b^{(k)}\boldsymbol{s}(\boldsymbol{z}^{(k)})]^T \boldsymbol{P}[\boldsymbol{z}^{(k)} - \boldsymbol{x}_0] \qquad (44)$$
$$+ \beta^{(k)2}\|a^{(k)}\nabla d(\boldsymbol{z}^{(k)} - \boldsymbol{x}_0) + b^{(k)}\boldsymbol{s}(\boldsymbol{z}^{(k)})\|_2^2$$
$$\le (1 - 2\beta^{(k)}\gamma)\|\boldsymbol{P}[\boldsymbol{z}^{(k)} - \boldsymbol{x}_0]\|_2^2 + \beta^{(k)2}B^2$$

The second equality is from Eq. (6). The last line is from assumption 5 and Eq. (31).

Next, combining Eqs. (43) and (44), we have

$$\|\boldsymbol{x}^{(k+1)} - P[\boldsymbol{x}^{(k+1)}]\|_2^2 \le \left(1 + \frac{2\alpha n L_s \kappa \beta^{(k)}}{m}\right)^2 (1 - 2\beta^{(k)}\gamma)\|\boldsymbol{x}^{(k)} - P[\boldsymbol{x}^{(k)}]\|_2^2 + \beta^{(k)2}D$$
$$\le (1 - 2\beta^{(k)}F)\|\boldsymbol{x}^{(k)} - P[\boldsymbol{x}^{(k)}]\|_2^2 + \beta^{(k)2}D$$

$$(45)$$

where

$$D = E + B^2$$
$$E = \frac{8\alpha\sqrt{n}L_s\kappa^2}{\delta m^2 r_1^{(0)2}} \qquad (46)$$
$$F = \gamma - \frac{2\alpha n L_s \kappa}{m} > 0 \text{ (Assumption 7)}$$

According to assumption 8, $\lim_{k\to\infty} \beta^{(k)} = 0$. Thus, $\forall \epsilon > 0$, $\exists K(\epsilon)$, $\forall k > K(\epsilon)$, we have $\beta^{(k)} \le 2\gamma\varepsilon/D$. Therefore

$$\|\boldsymbol{x}^{(k+1)} - P[\boldsymbol{x}^{(k+1)}]\|_2^2 - \epsilon \le (1 - 2F\beta^{(k)})(\|\boldsymbol{x}^{(k)} - P[\boldsymbol{x}^{(k)}]\|_2^2 - \epsilon) \qquad (47)$$

Finally, notice that by assumption 8

$$\lim_{K\to\infty} \prod_{k=1}^{K} (1 - 2\beta^{(k)}F) = 0 \qquad (48)$$

then

$$\lim_{k\to\infty} \|\boldsymbol{x}^{(k)} - P[\boldsymbol{x}^{(k)}]\|_2^2 \le \epsilon \qquad (49)$$

which holds $\forall \epsilon > 0$. This concludes the proof. $\qquad\square$

**Lemma 1.4.** *Given all the assumptions in Thm. 1, where $M_\alpha$ and $m_k$ defined in Eqs. (19) and (20), and*

$$M_\beta = \min\left\{1, \sqrt{\frac{C(1-A)}{B}}, \frac{2FX}{D}, \frac{1}{2F}, \sqrt{\frac{X - r_1^{(0)2}\|\boldsymbol{P}[\boldsymbol{x}^{(0)} - \boldsymbol{x}_0]\|_2^2}{E}}, \frac{X}{r_1^{(0)2}D/2F + E}\right\} \qquad (50)$$

*where A, B, D and E are defined in Eqs. (32) and (46), the following inequalities hold $\forall k$.*

$$|c(\boldsymbol{x}^{(k)})| \le C$$
$$|c(\boldsymbol{z}^{(k)})| \le C$$
$$\|\boldsymbol{P}[\boldsymbol{x}^{(k)} - \boldsymbol{x}_0]\|_2^2 \le X \qquad (51)$$
$$\|\boldsymbol{P}[\boldsymbol{z}^{(k)} - \boldsymbol{x}_0]\|_2^2 \le X$$

i.e. $\boldsymbol{x}^{(k)}, \boldsymbol{z}^{(k)} \in \mathbb{B}, \forall k$.

*Proof.* We will prove it by mathematical induction.

**Base Case:** From assumption 9, we have $|c(\boldsymbol{x}^{(0)})| \le C$ and $\|\boldsymbol{P}[\boldsymbol{x}^{(0)} - \boldsymbol{x}_0]\|_2^2 \le X$. Thus Eqs. (24) and (43) hold for $k = 0$. From Eq. (24) and assumption 7, we have $|c(\boldsymbol{z}^{(0)})| \le |c(\boldsymbol{x}^{(0)})| \le C$. From Eqs. (43) and (50), we have $\|\boldsymbol{P}[\boldsymbol{z}^{(0)} - \boldsymbol{x}_0]\|_2^2 \le X$.

**Step Case:** Assume Eq. (51) holds $\forall k \leq K$, then Eqs. (31) and (45) holds $\forall k \leq K$.

● Proving $|c(\boldsymbol{x}^{(K+1)})| \leq C$:

From Eq. (31),

$$|c(\boldsymbol{x}^{(K+1)})| - \frac{\beta^{(K)2}B}{1-A} \leq A\left(|c(\boldsymbol{x}^{(K)})| - \frac{\beta^{(K)2}B}{1-A}\right) \tag{52}$$

If

$$|c(\boldsymbol{x}^{(K)})| \leq \frac{\beta^{(K)2}B}{1-A}$$

Then

$$|c(\boldsymbol{x}^{(K+1)})| \leq \frac{\beta^{(K)2}B}{1-A} \leq \frac{\beta^{(0)2}B}{1-A} \leq C \tag{53}$$

where the last inequality is given by Eq. (50).

Otherwise

$$|c(\boldsymbol{x}^{(K+1)})| \leq |c(\boldsymbol{x}^{(K)})| \leq C \tag{54}$$

● Proving $\|\boldsymbol{P}[\boldsymbol{x}^{(K+1)} - \boldsymbol{x}_0]\|_2^2 \leq X$:

From Eq. (45)

$$\|\boldsymbol{P}[\boldsymbol{x}^{(K+1)} - \boldsymbol{x}_0]\|_2^2 - \frac{\beta^{(K)}D}{2F} \leq (1 - 2\beta^{(K)}F)\left(\|\boldsymbol{P}[\boldsymbol{x}^{(K)} - \boldsymbol{x}_0]\|_2^2 - \frac{\beta^{(K)}D}{2F}\right) \tag{55}$$

Notice that from Eq. (50), $0 \leq (1 - 2\beta^{(0)}F) \leq (1 - 2\beta^{(K)}F) < 1$.

If

$$\|\boldsymbol{P}[\boldsymbol{x}^{(K)} - \boldsymbol{x}_0]\|_2^2 \leq \frac{\beta^{(K)}D}{2F}$$

Then

$$\|\boldsymbol{P}[\boldsymbol{x}^{(K+1)} - \boldsymbol{x}_0]\|_2^2 \leq \frac{\beta^{(K)}D}{2F} \leq \frac{\beta^{(0)}D}{2F} \leq X \tag{56}$$

where the last inequality is given by Eq. (50).

Otherwise

$$\|\boldsymbol{P}[\boldsymbol{x}^{(K+1)} - \boldsymbol{x}_0]\|_2^2 \leq \|\boldsymbol{P}[\boldsymbol{x}^{(K)} - \boldsymbol{x}_0]\|_2^2 \leq X \tag{57}$$

● Proving $|c(\boldsymbol{z}^{(K+1)})| \leq C$:

Since we have established $|c(\boldsymbol{x}^{(K+1)})| \leq C$, Eq. 24 holds for $k = K + 1$. Therefore

$$|c(\boldsymbol{z}^{(K+1)})| = A|c(\boldsymbol{x}^{(K+1)})| \leq |c(\boldsymbol{x}^{(K+1)})| \leq C \tag{58}$$

● Proving $\|\boldsymbol{P}[\boldsymbol{z}^{(K+1)} - \boldsymbol{x}_0]\|_2^2 \leq X$:

Since we have established $\|\boldsymbol{P}[\boldsymbol{x}^{(K+1)} - \boldsymbol{x}_0]\|_2^2 \leq X$, Eq. (43) holds for $k = K + 1$.

From Eqs. (56) and (57), we can establish, through recursion, that

$$\|\boldsymbol{P}[\boldsymbol{x}^{(k)} - \boldsymbol{x}_0]\|_2^2 \leq \max\left\{\frac{\beta^{(0)}D}{2F}, \|\boldsymbol{P}[\boldsymbol{x}^{(0)} - \boldsymbol{x}_0]\|_2^2\right\}, \forall k \leq K + 1 \tag{59}$$

Therefore,

$$
\begin{aligned}
\|\boldsymbol{P}[\boldsymbol{z}^{(K+1)} - \boldsymbol{x}_0]\|_2^2 &\leq r_1^{(K+1)2}\|\boldsymbol{P}[\boldsymbol{x}^{(K+1)} - \boldsymbol{x}_0]\|_2^2 + \beta^{(K+1)2}E \\
&\leq r_1^{(K+1)2}\max\left\{\frac{\beta^{(0)}D}{2F}, \|\boldsymbol{P}[\boldsymbol{x}^{(0)} - \boldsymbol{x}_0]\|_2^2\right\} + \beta^{(K+1)2}E \\
&\leq r_1^{(0)2}\max\left\{\frac{\beta^{(0)}D}{2F}, \|\boldsymbol{P}[\boldsymbol{x}^{(0)} - \boldsymbol{x}_0]\|_2^2\right\} + \beta^{(0)2}E \\
&\leq X
\end{aligned}
\tag{60}
$$

The first line is given by Eq. (43). The last line is given by Eq. (50). $\qquad\square$

**Lemma 1.5.** *Under the assumptions in Thm. 1*

$$\boldsymbol{P}[\boldsymbol{x}^* - \boldsymbol{x}_0] = 0 \tag{61}$$

*Proof.* From Thm. 2, a solution, denoted as $\boldsymbol{x}'$, to

$$\min_{\boldsymbol{x}} d(\boldsymbol{x} - \boldsymbol{x}_0)$$
$$\text{s.t.} \nabla^T c(\boldsymbol{x}^*)(\boldsymbol{x} - \boldsymbol{x}^*) = 0 \tag{62}$$

would satisfy

$$\boldsymbol{P}[\boldsymbol{x}' - \boldsymbol{x}_0] = 0 \tag{63}$$

If $\boldsymbol{P}[\boldsymbol{x}^* - \boldsymbol{x}_0] \neq 0$, there are two possibilities. The first possibility is that $\boldsymbol{x}^*$ is not a solution to Eq. (62), which contradicts with the first order optimality condition that $\boldsymbol{x}^*$ must satisfy.

The second possibility is there are multiple solutions to the problem in Eq. (62), and $\boldsymbol{x}'$ and $\boldsymbol{x}^*$ are both its solutions. This can happen if $d(\cdot)$ is $\ell_1$ or $\ell_\infty$ norm. By definition

$$\nabla^T c(\boldsymbol{x}^*)(\boldsymbol{x}' - \boldsymbol{x}^*) = 0 \tag{64}$$

Since $\boldsymbol{x}^*$ is a local minimum to Eq. (1), $\exists j \in \mathcal{I}, \varepsilon < 1, \forall \delta < \varepsilon, \boldsymbol{x}_\delta = \delta \boldsymbol{x}^* + (1 - \delta)\boldsymbol{x}'$, s.t.

$$c(\boldsymbol{x}_\delta) > 0, \ \boldsymbol{x}_\delta \in \mathbb{B}_j \tag{65}$$

Otherwise, if $c(\boldsymbol{x}_\delta) \leq 0$, then $\boldsymbol{x}_\delta$ is a feasible solution to the problem in Eq. (1) and

$$d(\boldsymbol{x}_\delta - \boldsymbol{x}_0) \leq \delta d(\boldsymbol{x}^* - \boldsymbol{x}_0) + (1 - \delta)d(\boldsymbol{x}' - \boldsymbol{x}_0) = d(\boldsymbol{x}^* - \boldsymbol{x}_0) \tag{66}$$

which contradicts with the assumption that $\boldsymbol{x}^*$ is a unique local optimum in $\mathbb{B}$.

Eq. (65) implies

$$\nabla_j^T c(\boldsymbol{x}_\delta)(\boldsymbol{x}_\delta - \boldsymbol{x}^*) = (1 - \delta)\nabla_j^T c(\boldsymbol{x}_\delta)(\boldsymbol{x}' - \boldsymbol{x}^*) > 0 \tag{67}$$

On the other hand, notice that Eq. (63) implies $\boldsymbol{s}(\boldsymbol{x}^*) = \lambda(\boldsymbol{x}_0 - \boldsymbol{x}'), \lambda > 0$. For $\ell_1/\ell_\infty$ cases, $\boldsymbol{s}_j(\boldsymbol{x})$ takes discrete values. Therefore, to satisfy assumption 2, $\boldsymbol{s}_j(\boldsymbol{x}_\delta) = \boldsymbol{s}_j(\boldsymbol{x}^*)$, which implies

$$0 = d(\boldsymbol{x}^*) - d(\boldsymbol{x}^*) \geq -\left[\lambda_j \nabla^T c_j(\boldsymbol{x}_\delta) + \sum_{i \in I, i \neq j} \lambda_i \nabla^T c_i(\boldsymbol{x}^*)\right](\boldsymbol{x}^* - \boldsymbol{x}'), \lambda_i > 0, i \in \mathcal{I} \tag{68}$$

The first inequality is because $-\left[\lambda_j \nabla^T c_j(\boldsymbol{x}_\delta) + \sum_{i \in I, i \neq j} \lambda_i \nabla^T c_i(\boldsymbol{x}^*)\right] \in \nabla_s^T d(\boldsymbol{x}' - \boldsymbol{x}_0)$.

Eqs. (67) and (68) cause a contradiction. □

Now we are ready to prove Thm. 1.

*Proof of Thm. 1.* From Lems. 1.2, 1.3 and 1.4, we can established that Eqs. (21) and (37) holds under all the assumptions in Thm. 1. The only thing we need to prove is that Eqs. (21) and (37) necessarily implies $\lim_{k \to} \|\boldsymbol{x}^{(k)} - \boldsymbol{x}_0\|_2 = 0$.

First, from Lem. 1.5

$$\|\boldsymbol{P}[\boldsymbol{x}^* - \boldsymbol{x}_0]\|_2 = 0 \tag{69}$$

Then, $\forall \boldsymbol{x}' \in \mathbb{B}$ s.t. $\|\boldsymbol{P}[\boldsymbol{x}' - \boldsymbol{x}_0]\|_2^2 = 0$, we have $\boldsymbol{x}' - \boldsymbol{x} = \lambda \boldsymbol{s}(\boldsymbol{x}')$. From assumption 4, we know that $c(\boldsymbol{x}')$ is monotonic along $\boldsymbol{x}' - \boldsymbol{x} = \lambda \boldsymbol{s}(\boldsymbol{x}')$. Therefore, $\boldsymbol{x}^*$ is the only point in $\mathbb{B}$ that satisfies $\|\boldsymbol{P}[\boldsymbol{x}' - \boldsymbol{x}_0]\|_2^2 = 0$ and $c(\boldsymbol{x}') = 0$.

Also, notice that $\boldsymbol{P}[\boldsymbol{x} - \boldsymbol{x}_0]$ and $c(\boldsymbol{x})$ are both continuous mappings. This concludes the proof. □

**Theorem 2.** *The solution to*

$$\arg\min_{\boldsymbol{x}} d(\boldsymbol{x})$$
$$\text{s.t. } \nabla^T c(\boldsymbol{x}')\boldsymbol{x} = b \tag{70}$$

*is*

$$\boldsymbol{x} = \frac{b\boldsymbol{s}(\boldsymbol{x}')}{\nabla^T c(\boldsymbol{x}')\boldsymbol{s}(\boldsymbol{x}')} \tag{71}$$

*Proof.* Decompose $\boldsymbol{x} = \lambda \boldsymbol{y}$, where $d(\boldsymbol{y}) = 1$. Then Eq. (70) can be rewritten as

$$\underset{\lambda, \boldsymbol{y}:d(\boldsymbol{y})=1}{\arg\min} \ \lambda \tag{72}$$
$$\text{s.t. } \lambda \nabla^T c(\boldsymbol{x}')\boldsymbol{y} = b$$

Notice that the product of $\lambda$ and $\nabla^T c(\boldsymbol{x}')\boldsymbol{y}$ is constant, so if $\lambda$ is to be minimized, then $\nabla^T c(\boldsymbol{x}')\boldsymbol{y}$ needs to be maximized. Namely, $\boldsymbol{y}$ can be determined by solving

$$\underset{\boldsymbol{y}:d(\boldsymbol{y})=1}{\max} \ \nabla^T c(\boldsymbol{x}')\boldsymbol{y} \tag{73}$$

which is the definition of dual norm. Therefore

$$\boldsymbol{y} = \boldsymbol{s}(\boldsymbol{x}') \tag{74}$$

Plug Eq. (74) into the constraint in Eq. (72), we can solve for $\lambda$. This concludes the proof.

$\square$

As a remark, Thm. 2 is applicable to the optimization problems in Eqs. (3) and (62) by changing the variable $\tilde{\boldsymbol{x}} = \boldsymbol{x} - \boldsymbol{x}_0$ and redefining $b$ accordingly.

**Theorem 3.** *For $\ell_2$ norm, if all the assumptions in Thm. 1, but assumption 5, hold, and*

$$\exists m_a > 0, \ s.t. \ a^{(k)} \geq m_a \tag{75}$$

*also assuming $c(\boldsymbol{x})$ is convex in $\mathbb{B}$, then $\exists \mathbb{B}' = \{\boldsymbol{x} : \|\boldsymbol{P}[\boldsymbol{x} - \boldsymbol{x}^*]\|_2^2 \leq X', |c(\boldsymbol{x})| \leq C'\} \subset \mathbb{B}$, s.t. assumption 5 hold.*

*Proof.* Since $c(\boldsymbol{x})$ is convex in $\mathbb{B}$, we have $\forall \boldsymbol{x}$

$$(\nabla^T c(\boldsymbol{x}) - \nabla^T c(\boldsymbol{x}^*))(\boldsymbol{x} - \boldsymbol{x}^*) \geq 0 \tag{76}$$

Further, assume $\boldsymbol{x}$ satisfies

$$\nabla^T c(\boldsymbol{x}^*)(\boldsymbol{x} - \boldsymbol{x}^*) = 0 \tag{77}$$

Then we have

$$\boldsymbol{P}^T \boldsymbol{P}(\boldsymbol{x} - \boldsymbol{x}^*) = \boldsymbol{P}(\boldsymbol{x} - \boldsymbol{x}^*) = \boldsymbol{x} - \boldsymbol{x}^* \tag{78}$$

where the first equality is from the fact that $\boldsymbol{P}$ is an orthogonal projection matrix under $\ell_2$ norm; the second equality is from the fact that the projection subspace of $\boldsymbol{P}$ is orthogonal to $\nabla c(\boldsymbol{x}^*)$ by construction.

Also, from Lem. 1.5, we have

$$\boldsymbol{P}(\boldsymbol{x} - \boldsymbol{x}^*) = \boldsymbol{P}(\boldsymbol{x} - \boldsymbol{x}_0) \tag{79}$$

Plug Eqs. (77) to (79) into (76), we have

$$\nabla^T c(\boldsymbol{x})\boldsymbol{P}^T \boldsymbol{P}(\boldsymbol{x} - \boldsymbol{x}^*) \geq 0 \tag{80}$$

On the other hand, let $\gamma = m_a/2\|\boldsymbol{x}^* - \boldsymbol{x}_0\|_2$, then $\forall x$ satisfying Eq. (77) and $\boldsymbol{x} \neq \boldsymbol{x}^*$

$$
\begin{aligned}
a^{(k)}\nabla^T d(\boldsymbol{x} - \boldsymbol{x}_0)\boldsymbol{P}^T \boldsymbol{P}(\boldsymbol{x} - \boldsymbol{x}_0) &= \frac{a^{(k)}}{\|\boldsymbol{x} - \boldsymbol{x}_0\|_2}(\boldsymbol{x} - \boldsymbol{x}_0)^T \boldsymbol{P}^T \boldsymbol{P}(\boldsymbol{x} - \boldsymbol{x}_0) \\
&\geq \frac{m_a}{\|\boldsymbol{x}^* - \boldsymbol{x}_0\|_2}(\boldsymbol{x} - \boldsymbol{x}_0)^T \boldsymbol{P}^T \boldsymbol{P}(\boldsymbol{x} - \boldsymbol{x}_0) \\
&> \gamma(\boldsymbol{x} - \boldsymbol{x}_0)^T \boldsymbol{P}^T \boldsymbol{P}(\boldsymbol{x} - \boldsymbol{x}_0)
\end{aligned}
\tag{81}
$$

where the second line comes from Eq. (75) and the fact that $\boldsymbol{x}^*$ is the optimal solution to the problem in Eq. (62). Combining Eqs. (80) and (81), we know that assumption 5 holds *with strict inequality* for $\boldsymbol{x}$ satisfying Eq. (77) and $\boldsymbol{x} \neq \boldsymbol{x}^*$.

$\nabla^T c(\boldsymbol{x})\boldsymbol{P}^T \boldsymbol{P}(\boldsymbol{x} - \boldsymbol{x}^*)$, $\nabla^T d(\boldsymbol{x} - \boldsymbol{x}_0)\boldsymbol{P}^T \boldsymbol{P}(\boldsymbol{x} - \boldsymbol{x}_0)$ and $\gamma(\boldsymbol{x} - \boldsymbol{x}_0)^T \boldsymbol{P}^T \boldsymbol{P}(\boldsymbol{x} - \boldsymbol{x}_0)$ are continuous functions, and therefore $\exists \mathbb{B}'$ where assumption 5 also holds. This concludes the proof.

$\square$

## B  HYPERPARAMETER SETTINGS FOR MARGINATTACK

Table 4 list the hyperparameter settings for MARGINATTACK.

Table 4: Hyperparameter settings for MARGINATTACK.

| Hyper-parameters | $\ell_2$ | | | $\ell_\infty$ | | |
|---|---|---|---|---|---|---|
| | MNIST | CIFAR | IMAGENET | MNIST | CIFAR0 | IMAGENET |
| $a^{(k)}$ | Eq. (8) | Eq. (8) | Eq. (8) | 0.1 | 1 | 0.3 |
| $b^{(k)}$ | 1 | 1 | 1 | Eq. (10) | Eq. (10) | Eq. (10) |
| $\alpha$ | 1 | 1 | 1 | 0.2 | 0.2 | 0.2 |
| $\beta^{(k)}$ | $(k+1)^{-0.5}$ | $(k+1)^{-0.5}$ | $(k+1)^{-0.5}$ | $(k+1)^{-1}$ | $(k+1)^{-1}$ | $(k+1)^{-1}$ |
| $u$ | 0.05 | 0.1 | 0 | 0.05 | 0.1 | 0 |
| $\varepsilon$ | -0.01 | | | | | |

