# OpenReview forum: "An Efficient and Margin-Approaching Zero-Confidence Adversarial Attack"
_ICLR.cc/2019/Conference_

### Official Review · AnonReviewer1 · 2018-11-02
**Unclear problem statement; mixed results**

**Rating:** 6
**Confidence:** 5

**Review:**

i have change my rating from 5 to 6 after reading the numerous and thorough rebuttals from the authors. I hope they will incorporate these clarifications and additional experiments into the final version of the paper if accepted.

The purpose of this paper is presumably to approximate the margin of a sample as accurately as possible. This is clearly an intractable problem. Thus all attacks make some kind of approximation, including this paper. I am still a bit confused about the difference between "zero-confidence attacks" and those that don't fall into that category such as PGD. Since all of these are approximations, and we cannot know how far we are from the true margin, I don't see why these categories help. The authors spend two paragraphs in the introduction trying to draw a distinction but I am still not convinced.

The proofs provided by the authors assume that convexity and many assumptions, which makes it not very useful for the real world case. What would have been helpful is to show the accuracy of their margin for simple binary toy 2D problems, where the true margin and their approximation can be visualized. This was not done. This reduces the paper to an empirical exercise rather than a true understanding of their method's advantages and limitations.

Finally, the experimental results do not show any significant advantage over PGD, either in running time (they are slower) or norm perturbation. Thus their novelty rests on the definition of zero confidence attack, and of the importance of such a attack. So clarifying the above question will help to judge the paper's novelty.

---

> ### Author Response · Authors · 2018-11-14
> **Zero-Confidence vs Fix-Perturbation**
>
> The following link is a figure that explains the difference between zero-confidence attack and fix-perturbation attack.
>
> https://docs.google.com/viewer?url=https://raw.githubusercontent.com/anon181018/iclr2019_rebuttal/master/figure1.pdf
>
> As can be seen, the zero-confidence attack finds the closest point on the decision boundary; while fix perturbation-attack finds adversarial samples within a fix perturbation. Both attacks are equivalent if we only want to compute the attack success rate under a given perturbation level. However, we will be better off with zero-confidence attacks if we want to
>
> 1) Compute the margin of each individual example; and
> 2) Probe and study the decision boundary of a classifier
>
> Of course, we can also measure the margin of each example using a fix-perturbation attack, for example PGD, by binary searching over the perturbation levels. However, the computation cost will significantly increase. Consider, for example, the CIFAR-10 dataset. Since for our model, most margins fall within 10, so let’s assume the binary search range is 10 (for adversarially trained models this number will be much higher). If we want to achieve a accuracy of 0.1, then we need at least 7 binary search steps. In other words, the computation complexity increases by 7 times. In fact, CW applies a similar binary search idea to achieve zero-confidence attack, and that is why its computation cost is high. The above discussion is not saying that it is impossible to convert PGD to a zero-confidence attack efficiently, but it at least provides a perspective on why zero-confidence attack is challenging, and why the complexity reduction as well as accuracy improvement of MarginAttack is valuable.

---

> > ### Comment · AnonReviewer1 · 2018-11-19
> > **Thanks - reading through these**
> >
> > Thank you for posting all these detailed results and explanations. I have a much better understanding of the motivations and am reading through all your responses at present.

---

> ### Author Response · Authors · 2018-11-14
> **Regarding the theorem assumptions**
>
> Although this is not the major focus of your comment, we would like to revisit the theorem assumptions. While there are nine assumptions, these assumptions are in fact more realistic than expected. Take the convexity assumption, which you mentioned in your review, as an example. This assumption does not say that the constraint has to be convex. It only says that the constraint should not be ‘too concave’. In particular, the curvature of the of the decision should not exceed that of the L2 or L-infinity ball. For better illustration, we plotted some decision boundaries that are allowed by the assumption, and some that are not.
>
> Please check the following link:
>
> https://docs.google.com/viewer?url=https://raw.githubusercontent.com/anon181018/iclr2019_rebuttal/master/figure2.pdf
>
> As can be seen, the convexity assumption permits a wide variety of decision boundaries. Among the few cases that it does not permit is the case where the decision boundary bends more than the L2 ball does. In this case, the critical point becomes a local maximum rather than a local minimum.
>
> The other assumptions are also more realistic than their names sound. The differentiability assumption does not stipulate that the constraint has to be differentiable. It actually permits countably infinite jump discontinuities. The Lipchitz continuous assumption does not assume Lipchitz continuity everywhere, but only at x*. We are not saying that the assumptions are very loose, but they are realistic enough to shed some light on the actual convergence property of MarginAttack. Nevertheless, we are considering adding a 2D toy example as you suggested. We will post further responses if there are further updates.

---

### Official Review · AnonReviewer3 · 2018-11-06
**Claims to be significantly faster than the CW attack, but I have some questions about the experiments**

**Rating:** 5
**Confidence:** 4

**Review:**

The authors propose a new method for constructing adversarial examples called MarginAttack. The method is inspired by Rosen's algorithm, a classical algorithm in constrained optimization. At its core, Rosen's algorithm (instantiated for adversarial examples) alternates between moving towards the set of misclassified points and moving towards the original data point (while ensuring that we do not move too far away from the set of misclassified points). The authors provide theoretical guarantees (local convergence) and a broad set of experiments. The experiments show that MarginAttack finds adversarial examples with small distortion (as good as the baselines or slightly better), and that the algorithm runs faster than the Carlini-Wagner (CW) baseline (but slower than other methods).

The authors make a distinction between "fixed perturbation" attacks and "zero confidence" attacks. The former finds the strongest attack within a given constrained set, while the latter finds the smallest perturbation that leads to a misclassification. Method such as projected gradient descent fall into the "fixed perturbation" category, while MarginAttack and CW belong to the "zero confidence" category. The authors claim that zero confidence attacks pose a harder problem and hence mainly compare their experimental results to the CW attack. Indeed, their results show that MarginAttack is 3x - 5x faster than CW and sometimes achieves smaller perturbations.

First of all, I would like to emphasize that the authors conducted a thorough experimental study on multiple datasets using multiple baseline algorithms. Unfortunately, the comparison to CW and PGD still leaves some questions in my opinion:

- The authors state that CW does an internal binary search over the Lagrangian multiplier, and that this search goes for up to 10 steps. As a result, it is not clear whether the running time benchmarks are a fair comparison since MarginAttack does not automatically tune its parameters. To the best of my knowledge, the CW implementation in Cleverhans is specifically set up so that the user does not need to tune a large number of hyperparameters (the implementation accepts a running time overhead to achieve this). Since MarginAttack also contains multiple hyperparameters (see Table 4), it would be interesting to see how the running time of MarginAttack compares to that of a tuned CW implementation without the binary search.

- The authors explicitly state that the step sizes for CW were tuned for best performance, but do not mention this for PGD. For a fair comparison, the step sizes used for PGD should also be (approximately) tuned. Moreover, it is not clear why PGD is only used for an l_inf comparison and not a l_2 comparison.

- In the introduction, the authors emphasize the distinction between fixed perturbation attacks and zero confidence attacks. However, from an optimization point of view, these two notions are clearly related and a fixed perturbation attack can be converted to a small perturbation / zero confidence attack via a binary search over the perturbation size. While one would indeed expect an overhead due to the binary search, it is not clear a priori how large this overhead needs to be to achieve a competitive zero confidence attack with PGD (especially with a tuned step size for PGD, see above).

I would be grateful if the authors could provide their view on these points. Until then, I will assign a rating of 5 since tuning the parameters of optimization algorithms is crucial for a fair comparison.


Additional comments:

- In the introduction, the authors equate white-box attacks with access to gradient information. But generally a white-box attack is understood as an attack that has arbitrary access to the target network. It may be helpful for the reader to clarify this.

- In the second paragraph of the introduction, the authors claim that fixed perturbation attacks and zero confidence attacks differ significantly. But as pointed out above, it is possible to convert a fixed perturbation attack to a zero confidence attack via a binary search. So it is not clear that there is a large gap in difficulty. Moreover, the authors state that fixed perturbation attacks often come with theoretical guarantees. But to the best of my knowledge, there is no comprehensive theory that describes when a fixed perturbation attack should be expected to succeed in attacking a commonly used neural network.

- On top of Page 2, the authors claim that zero-confidence attacks are a more realistic attack setting. Why is that?

- The authors state that JSMA (Papernot et al., 2016) is one of the earliest works that use gradient information for constructing adversarial examples. However, L-BFGS as employed by Szegedy et al., 2013 also uses gradient information. Moreover, the authors may want to cite the work of Biggio et al. from 2013 (see the survey https://arxiv.org/abs/1712.03141).

- Since all distances referred to by d(x, y) seem to be norms (and the paper relies on the existence of dual norms), it may be more clear for the reader to use the norm notation || . || from the beginning.

---

> ### Author Response · Authors · 2018-11-14
> **Regarding your other reviews**
>
> First, regarding the white box attack definition. Yes, the white box attack is understood as having access to all network information, including structure and parameters. So it is possible to compute gradient information. Black box attacks only have access to logits or only decision, so it is not possible to accurately compute the gradient information. We will make this distinction clearer.
>
> Second, the PGD convergence guarantee we meant is only about local convergence. Under mild assumptions, PGD is able to converge to a critical point of the PGD loss function, where no feasible direction can increase the loss function. We will clarify this in our updated version.
>
> Third, by a ‘more realistic attack’, we meant that under a true attack setting, an attacker would not confine himself to a fix perturbation, but is more likely to keep attacking until success, while minimizing perturbation.
>
> Fourth, we will correct our statement about the earliest work that incorporates gradient information into adversarial attack.
>
> Finally, we will change the norm notation.

---

> ### Author Response · Authors · 2018-11-14
> **Regarding the comparison with PGD**
>
> First we would like to clarify that the learning rate of PGD is tuned the same way as for CW. We somehow missed this statement in the paper. We will add this statement back to the paper.
>
> Second, yes it is entirely possible to convert PGD to a zero-confident attack. In our response to another reviewer, we estimated the computational overhead. We will copy the analysis here. Consider, for example, the CIFAR-10 dataset. Since for our model, most margins fall within 10, so let’s assume the binary search range is 10 (for adversarially trained models this number will be much higher). If we want to achieve an accuracy of 0.1, then we need at least 7 binary search steps. In other words, the computation complexity increases by 7 times. The above discussion is not saying that it is impossible to convert PGD to a zero-confidence attack efficiently, but it at least provides a perspective on why the complexity reduction as well as accuracy improvement of MarginAttack is valuable.
>
> Finally, we would like to point out that while PGD is a state-of-the-art in L-infinity attack. It is not in L2 attack. One of the reasons is that PGD alternatively projects onto the constraint box and L2 ball, which is not equivalent to projecting onto the intersect of both. In L-infinity attack they are equivalent. The following link is a figure that provides an illustration on this.
>
> https://docs.google.com/viewer?url=https://raw.githubusercontent.com/anon181018/iclr2019_rebuttal/master/figure3.pdf
>
> That is the reason why we did not incorporate PGD L2 in our comparison. However, we would like to provide the results here.
>
> MNIST:
>
> Perturb. Lev.	MarginAttack	PGD L2
> 1		        25.69			12.53
> 1.41		        66.34			37.11
> 1.73		        88.40			64.13
> 2		        97.11			81.68
>
> CIFAR10:
>
> Perturb. Lev.	MarginAttack	PGD L2
> 8		        24.27			12.94
> 15		        46.37			25.30
> 25		        73.82			46.90
> 40		        93.29			57.13
>
> IMAGENET:
>
> Perturb. Lev.	MarginAttack	PGD L2
> 10		        40.42			29.45
> 32		        60.59			40.73
> 50		        74.89			50.80
> 80		        89.73			66.35
>
> Hope the above clarifications help.

---

> ### Author Response · Authors · 2018-11-14
> **Regarding comparison with CW attack**
>
> Thank you for bringing up the binary search issue. We would like to clarify that the binary search is an integral part of the CW attack and that it cannot be replaced with hyperparameter tuning beforehand. This is because the purpose of the binary search is to find the Lagrange multiplier for the Lagrangian, which is specific to *each individual token*. In other words, each different input sample comes with a different optimal Lagrange multiplier. Therefore, it is impossible to tune a universal Lagrange multiplier and remove the binary search. The CW algorithm can be regarded as a two-way optimization problem. For each sample, it first optimizes over the Lagrange multiplier via binary search, and then optimizes over the adversarial sample via gradient descent. In short, the Lagrange multiplier is technically not a hyperparameter, but an optimization variable just like the adversarial sample itself.
>
> The CW implementation does come with a set of hyperparameters that it asks the users to tune, including the initial Lagrange multiplier guess and the initial step size, both of which are already tuned to its best performance.
>
> Nevertheless, although the binary search cannot be removed, we are interested to see what will happen is it is reduced. For this we perform an additional experiment where the number of binary search steps is reduced to 5 (named CW bin5), 3 (named CW bin3) and 1(named CW bin1) on MNIST and CIFAR10. Below are the attack success rates under different perturbation levels.
>
> MNIST:
>
> Perturb. Lev.	MarginAttack	CW bin10	CW bin5	        CW bin3	       CW bin1
> 1		        25.69			24.86		24.82		24.63		9.94
> 1.41		        66.34			63.23		63.11		62.72		9.98
> 1.73		        88.40			85.94		85.90		85.66		9.99
> 2		        97.11			95.42		95.36		95.25		9.99
>
> CIFAR10:
>
> Perturb. Lev.	MarginAttack	CW bin10	CW bin5	        CW bin3
> 8		        24.27			24.11		24.02		9.93
> 15		        46.37			45.52		45.22		13.95
> 25		        73.82			71.76		70.92		22.26
> 40		        93.29			91.61		90.65		34.48
>
> Below is the attack time.
>
> MNIST:
>
> MarginAttack	CW bin10	CW bin5	CW bin3	CW bin1
> 3.01			        16.02		8.99		5.77		1.37
>
> CIFAR10:
>
> MarginAttack	CW bin10	CW bin5	        CW bin3
> 51.03			234.75		102.68		33.98
>
> As can be seen, the performance does drop as the number binary search steps decreases. In particular, the algorithm completely fails when the binary search step number drops below a certain threshold (1 for MNIST and 3 for CIFAR10). Upon failure threshold, there is an unproportional drop in running time, which is probably due to the early stop mechanism in CW. We conjecture that the threshold is higher when the dataset has greater variations. These results provide more complete evidence on how MarginAttack is able to achieve a much better accuracy-efficiency tradeoff than CW. We will add these results to the paper.
>
> Hope this clarification helps.

---

> > ### Public Comment · ~Nicholas_Carlini1 · 2018-11-14
> > **Selecting the number of binary search steps**
> >
> > Just a quick comment on selecting the number of binary search steps: if you make your initial guess reasonable, then usually you don't need more than two or three steps. I have no theoretical basis for these numbers, but 1.0 for MNIST and 0.1 for CIFAR typically works well when using a range of [0,1] for the input image.
> >
> > If you're using [0,255] then you'll want to change your initial guess to be something that fits that range.

---

> ### Author Response · Authors · 2018-11-19
> **Updated results on CiFAR10**
>
> Thanks to the hyperparameter tuning suggestions, we are able to achieve a set of better results on CiFAR10. The updated results are as follows:
>
> Attack success rate:
>
> Perturb. Lev.	MarginAttack	CW bin10	CW bin5	        CW bin3	       CW bin1
> 8		        24.27			24.04		23.99		23.89             15.14
> 15		        46.37			45.56		45.39		45.21             20.53
> 25		        73.82			71.80		71.66		71.56             20.57
> 40		        93.29			92.10		91.86		91.49             20.57
>
> Running time:
> MarginAttack	CW bin10	CW bin5	        CW bin3	        CW bin1
> 51.03			350.10		168.88		100.10             24.30
>
> Compared to the previous results posted, the attack success rate of CWbin10 is almost the same, but the results of CW with fewer binary steps are improved. The basic conclusions do not change though. Notice that increasing the number of binary search steps does help to improve the success rate, but even compared with 10 binary search steps, MarginAttack still maintains a higher success rate at all levels. In the meantime, MarginAttack has a much lower running time, and thus strikes a better success-rate-balance-tradeoff.

---

> > ### Comment · AnonReviewer3 · 2018-12-01
> > **Further tuning?**
> >
> > I would like to thank the authors for running this second set of experiments.
> >
> > Did you also re-tune the step sizes after changing the binary search in the CW attack?

---

> > > ### Author Response · Authors · 2018-12-03
> > > **Yes**
> > >
> > > Yes. Each binary search step setting comes with a separate step size tuning.

---

> > > > ### Comment · AnonReviewer3 · 2018-12-03
> > > > **Two more questions**
> > > >
> > > > That is good to know, thank you.
> > > >
> > > > I have two more questions:
> > > >
> > > > - Do the CIFAR-10 results mentioned in this sub-thread already include the improved CW baseline mentioned in your comment from Dec 3rd "Updated CW results on ImageNet"?
> > > >
> > > > - How are the running times determined in your comparison above? Both MarginAttack and CW are iterative methods that can (in principle) be stopped early to improve the running time. So ideally, the two algorithms would be compared in a plot of running time vs attack success rate (with intermediate results from the algorithms after each iteration giving a trade-off curve).

---

> > > > > ### Author Response · Authors · 2018-12-03
> > > > > **Regarding updated results and running time comparison**
> > > > >
> > > > > Thank you for following up. Regarding your two questions
> > > > >
> > > > > 1. The updated results in this sub-thread are on CIFAR-10 and MNIST. The Dec 3rd comment is on ImageNet. Previously there have been discussions on the ImageNet results in the thread entitled 'Minor comments'. The Dec 3rd is the general response to these discussions.
> > > > >
> > > > > 2. Here are more details on the runtime settings of both algorithms. CW comes with an option 'abort_early' and it has been turned on. This option will abort the iterations when the algorithm converges. This option will accelerate the algorithm without hurting the performance. On the other hand, we didn't implement a similar mechanism in MarginAttack, so MarginAttack will run all the way through the end even if it already converges. This places an advantage on CW. In spite of this, as shown in the results in this thread, MarginAttack still runs faster and more accurate than most of the settings in CW.
> > > > >
> > > > > If the above discussion is not cogent enough, we have some results where the number of iteration is cut down to 200 (1/10 of the original number of iterations; binary search step set to 10):
> > > > >
> > > > > MNIST:
> > > > >
> > > > > Perturb. Lev.	Margin2000	        CW2000  	Margin200    CW200
> > > > > 1		        25.69			24.86		25.17		23.87
> > > > > 1.41		        66.34			63.23		64.99		60.60
> > > > > 1.73		        88.40			85.94		87.28		84.25
> > > > > 2		        97.11			95.42		96.51		94.11
> > > > >
> > > > > CIFAR10:
> > > > >
> > > > > Perturb. Lev.	Margin2000   	CW2000  	Margin200     CW200
> > > > > 8		        24.27			24.04		24.31		23.86
> > > > > 15		        46.37			45.56		45.63		44.98
> > > > > 25		        73.82			71.80		71.91		70.95
> > > > > 40		        93.29			92.10		91.71		91.02
> > > > >
> > > > > As can be seen, even Margin200 can outperform CW2000 in most of the scenarios (except for CIFAR10 perturbation level 40). Hope this is cogent enough to show the improved accuracy-efficiency tradeoff of MarginAttack.

---

### Official Review · AnonReviewer4 · 2018-11-13
**I cannot see why the proposed method is better than CW attack**

**Rating:** 5
**Confidence:** 3

**Review:**

This paper proposes an efficient zero-confidence attack algorithm, MARGINATTACK, which uses the modified Rosen's algorithm to optimize the same objective as CW attack. Under a set of conditions, the authors proved convergence of the proposed attack algorithm. My main concern about this paper is why this algorithm has a better performance than CW attack? I would suggest comparing with CW attack under different sets of hyper-parameters.

Minor comment:
The theoretical proof depends on the convexity assumption, I would also suggest comparing the proposed attack with CW and other benchmarks on some simple models that satisfy the assumptions.

---

> ### Public Comment · ~Nicholas_Carlini1 · 2018-11-14
> **Working on figuring out what's different**
>
> This review seems to focus mainly on one graph in Figure 3. While I agree that this is confusing, I suspect that it's just a problem with how the authors are running the attack on ImageNet. The proposed attack still does better than DeepFool by the same margin as on CIFAR-10 and MNIST. This is supported by the reproduction script from ( https://github.com/tensorflow/cleverhans/issues/813 ), which produces an improved curve that is nearly identical to the authors attack.
>
> So I probably wouldn't look unfavorably on the paper for that reason alone. This is something that can be probably explained and/or fixed. I'm trying to look at the code that's provided on the github issue to see if I can figure out what's going on.
>
> [I am intentionally avoiding commenting on the content of the paper in other way.]

---

> ### Author Response · Authors · 2018-11-19
> **Comparison with CW & We did not assume convexity**
>
> Regarding your first concern on the comparison with CW: In short, MarginAttack is able to achieve a higher attack success rate than CW AND a shorter running time. The paper may not make this point obvious enough probably because the curves are too thick to reveal the difference. To show this point clearly, we would like to refer you to the results in our response to reviewer 3, where we scanned through the number of binary search steps and measure the success rate and running time.
>
> As can be seen, MarginAttack has a higher success rate than all the versions of CW. There is a success-rate-efficiency tradeoff in CW, as a smaller binary search step number leads to a lower success rate. However, even with 10 binary search steps, CW is still unable to outperform MarginAttack in terms of success rate. On the other hand, with very small numbers of binary search steps, CW still runs slower than MarginAttack. Hope these results will clarify your major concern.
>
> Regarding your minor concern:
>
> In the theorem, we did not assume convexity. The assumption with the name 'convexity' is saying that the constraint set should not be 'too concave'. Please check the following figure where we listed what decision boundaries are permitted by our theorem and what not.
>
> https://docs.google.com/viewer?url=https://raw.githubusercontent.com/anon181018/iclr2019_rebuttal/master/figure2.pdf
>
> As can be seen, the convexity assumption permits a wide variety of decision boundaries. Among the few cases that it does not permit is the case where the decision boundary bends more than the L2 ball does. In this case, the critical point becomes a local maximum rather than a local minimum.

---

### Public Comment · (anonymous) · 2018-10-11
**Minor comments**

This is a very nice attack. It appears simple to implement and is formally well-justified.

A few minor questions and comments

The attack approach seems related to the Decision Attack (https://openreview.net/forum?id=SyZI0GWCZ from ICLR last year), am I right in this understanding?

Do you know why CW performs worse than DeepFool on Imagenet (Figure 3 upper right)? The Decision Attack paper finds that CW performs better than DeepFool on ImageNet (as does the original CW attack paper).

Table 3 is mildly deceptive that the the "Ours" row is on the bottom but it is not the fastest, whereas in Table 2 it is in the correct (ordered) position.

---

> ### Author Response · Authors · 2018-10-12
> **Regarding your comments**
>
> Thank you for your interests in our work! Regarding your comments:
>
> 1. Yes, the idea of 'crawling along the decision boundary' is related to the L2 version of MarginAttack (Eq. (8)), and serves as a good reference if a black-box version of MarginAttack is to be developed. So we will add this paper to our reference. Partly because it is a white-box attack, MarginAttack does not have to wait until it reaches the decision boundary before it moves along the decision boundary, which is shown to significantly improve convergence both empirically and theoretically. Also, MarginAttack encompasses much richer attack schemes, because the L-infinity version of MarginAttack (Eq. (10)), as well as other valid settings of a_k and b_k, follows a different projection move direction from along the decision boundary. Nevertheless, we appreciate that you point out this relevant paper, and we will update our reference list accordingly.
>
> 2. In fact, the decision attack paper also finds that CW performs worse than DeepFool on ImageNet. In the table at the bottom of page 6, CW gets larger median perturbation norms than DeepFool does for all of the three architectures on ImageNet. In particular, for the ResNet50 architecture (which we also used), the median perturbation norm of CW is 2.2e-7, and that of DeepFool is 7.5e-8.
>
> The original paper of CW attack may shed some light on this. According to the paper, CW does perform better than DeepFool on ImageNet (Table V), but that is for the best case only, which refers to choosing 100 randomly chosen adversarial classes to perform the targeted attack, and then finding the easiest case. We can also try this for CW, but considering the computation cost of CW is so high already, multiplying it by 100 would really make this accuracy-efficiency tradeoff not worthwhile. On the other hand, as our paper intends to show, MarginAttack does a much better accuracy-efficiency tradeoff.
>
> 3. Thank you for pointing out the ordering issue in table 3. It is not meant to create any false perceptions -- the differences in the numbers are quite distinct. But you are right, for better formality, we will adjust it in our updated version.
>
> Thank you again for your comments!

---

> > ### Public Comment · (anonymous) · 2018-10-12
> > **Still unsure about CW ImageNet results**
> >
> > Thank you for explaining the difference.
> >
> > It still surprises me that 100+ iterations of gradient descent with CW will do worse than DeepFool. Are you performing CW untargeted, to compare to DeepFool? On ImageNet with a ResNet50 I can reach 95% adversarial success with a L2 distortion of 150 (when on a scale of 0-255) with 100 iterations of CW. Can you clarify how you are getting "2.2e-7"? The numbers in the paper are between 0 and 200.
> >
> > Sorry to be picky about this, but given that your curve matches CW exactly for MNIST and CIFAR and is significantly better on ImageNet, I would like to make sure this is accurate.

---

> > > ### Author Response · Authors · 2018-10-12
> > > **CW settings**
> > >
> > > No worries. We appreciate your comments and would like to ensure our results are right.
> > >
> > > Before we discuss the CW settings, we would like to first clarify that the '2.2e-7' I mentioned is not our results. It is the results reported in the decision attack paper (https://openreview.net/forum?id=SyZI0GWCZ from ICLR last year) that you previously mentioned. They also find CW performs worse than DeepFool. According to my understanding and guess, this should be the per-pixel squared distance and the pixel range is [0, 1]. Therefore converting it to the regular L2 distortion should yield around 50. Again, this is only our interpretation of their results. We would need to consult the authors of that paper to confirm it.
> > >
> > > We are interested in your CW results and would like to know more about how you configured the attack. Did you use CleverHans? If yes, how did you set the configuration parameters?
> > >
> > > To be transparent, here are our CleverHans settings for CW:
> > >
> > > cw_params = {'binary_search_steps': 10,
> > >                          'y': l,
> > >                          'max_iterations': 2000,
> > >                          'learning_rate': 0.01 (also tried 0.001, 0.05 and 0.1),
> > >                          'batch_size': 100,
> > >                          'initial_const': 0.1 (also tried 0.01),
> > >                          'clip_min': 0,
> > >                          'clip_max': 255,
> > >                          'abort_early': False (also tried True)}
> > >
> > > We scanned over the candidate settings and the results we reported were the best. Please let us know if you find anything problematic. We will be happy to make it right.

---

> > > > ### Public Comment · (anonymous) · 2018-10-12
> > > > **... one more thing**
> > > >
> > > > One final check: did you run CW on the pre-softmax logits of the ResNet? This takes a few extra lines of code for Keras to do, because by default the ResNet50 gives a post-softmax probability distribution. You will have to remove the softmax operation from the resnet, or as a quick check you can add a tf.log() around the output of the model.
> > > >
> > > > CW is known to perform very poorly on the post-softmax probability outputs. On my code, if I incorrectly use the post-softmax probability values instead, I can reproduce similar (higher distortion) results to yours and the prior paper.

---

> > > > > ### Author Response · Authors · 2018-10-16
> > > > > **We used pre-softmax logits**
> > > > >
> > > > > Thanks for the reminder! Upon checking we confirm that we used the pre-softmax logits.
> > > > >
> > > > > We will keep on our efforts to ensure the results are right. If you would like to share your configuration details at any time, it is always welcomed. Thanks again for initiating the discussion!

---

> > > > > > ### Public Comment · (anonymous) · 2018-10-17
> > > > > > **Releasing Code for CW ImageNet**
> > > > > >
> > > > > > Raised a github issue on CleverHans to see what's going wrong. My code is given there which does 2-3x better than your figure on ImageNet.
> > > > > >
> > > > > > https://github.com/tensorflow/cleverhans/issues/813
> > > > > >
> > > > > > It would be helpful to see your code if you could share it anonymously.

---

> > > > > > > ### Author Response · Authors · 2018-10-18
> > > > > > > **Code posted**
> > > > > > >
> > > > > > > Just posted the code in the issue.

---

### Author Response · Authors · 2018-12-03
**Updated CW results on ImageNet**

With the help of the useful discussions in https://github.com/tensorflow/cleverhans/issues/813, we are able to get the CW ImageNet results right. We would like to update the results as follows:

Perturb. Lev.	MarginAttack	CW bin5
10		        40.42			40.36
32		        60.59			58.71
50		        74.89			70.99
80		        89.43			85.64

This table and a continuous curve will replace the original results in the paper.

---

### Meta-Review · Area_Chair1 · 2018-12-17
**Many questions - not convincing enough at this time**

**Confidence:** 4
**Recommendation:** Reject

**Metareview:**

The paper proposes a new method for adversarial attacks, MarginAttack, which finds adversarial examples with small distortion and runs faster than the CW baseline, but slower than other methods. The authors provide theoretical guarantees and a broad set of experiments.

In the discussion, a consistent concern has been that, experimentally, the method does not perform noticeably better than previous approaches. The authors mention that the lines are too thick to reveal the difference. It has been pointed out that this might be related to the way the experiments are conducted, but the proposed method still does better than other methods. AnonReviewer1 mentions that the assumptions needed for the theoretical part might be too strong, meaning that the main contribution of the paper is in the experimental side.

The comparisons with other methods and the assumptions made in the theorems seem to have caused quite some confusion and there was a fair amount of discussion. Following the discussion session, AnonReviewer1 updated his rating from 5 to 6 with high confidence.

The referees all rate the paper as not very strong, with one marginally above acceptance threshold and two marginally below the acceptance threshold.

Although the paper seems to propose valuable ideas, and it appears that the discussion has clarified many questions from the initial submission, the paper has not provided a clear, convincing, selling point at this time.